# Exact Bayesian Inference on Discrete Models via Probability Generating Functions: A Probabilistic Programming Approach

**Fabian Zaiser**
Department of Computer Science
University of Oxford, UK
`fabian.zaiser@cs.ox.ac.uk`

**Andrzej S. Murawski**
Department of Computer Science
University of Oxford, UK
`andrzej.murawski@cs.ox.ac.uk`

**C.-H. Luke Ong**
School of Computer Science and Engineering
Nanyang Technological University, Singapore
`luke.ong@ntu.edu.sg`

## Abstract

We present an exact Bayesian inference method for discrete statistical models, which can find exact solutions to a large class of discrete inference problems, even with infinite support and continuous priors. To express such models, we introduce a probabilistic programming language that supports discrete and continuous sampling, discrete observations, affine functions, (stochastic) branching, and conditioning on discrete events. Our key tool is *probability generating functions*: they provide a compact closed-form representation of distributions that are definable by programs, thus enabling the exact computation of posterior probabilities, expectation, variance, and higher moments. Our inference method is provably correct and fully automated in a tool called *Genfer*, which uses automatic differentiation (specifically, Taylor polynomials), but does not require computer algebra. Our experiments show that Genfer is often faster than the existing exact inference tools PSI, Dice, and Prodigy. On a range of real-world inference problems that none of these exact tools can solve, Genfer's performance is competitive with approximate Monte Carlo methods, while avoiding approximation errors.

## 1 Introduction

Bayesian statistics is a highly successful framework for reasoning under uncertainty that has found widespread use in a variety of fields, such as AI/machine learning, medicine and healthcare, finance and risk management, social sciences, climate science, astrophysics, and many other disciplines. At its core is the idea of representing uncertainty as probability, and updating prior beliefs based on observed data via Bayes' law, to arrive at posterior beliefs. A key challenge in Bayesian statistics is computing this posterior distribution: analytical solutions are usually impossible or intractable, which necessitates the use of approximate methods, such as Markov-chain Monte Carlo (MCMC) or variational inference. In this work, we identify a large class of discrete models for which exact inference is in fact possible, in particular time series models of count data, such as autoregressive models, hidden Markov models, switchpoint models. We achieve this by leveraging ***probability generating functions (GFs)*** as a representation of distributions. The GF of a random variable $X$ is defined to be the function $G(x) := \mathbb{E}[x^X]$. In probability theory, it is a well-known tool to study random variables and their distributions, related to the moment generating and the characteristic functions [14, Chapter 4]. In computer science, GFs have previously been used for the analysis of probabilistic programs [17, 4] and for exact inference on certain classes of graphical models [26, 27].

37th Conference on Neural Information Processing Systems (NeurIPS 2023).

Here we apply them uniformly in a much more general context via probabilistic programming, enabling exact inference on more expressive Bayesian models.

We characterize the class of supported models with the help of a probabilistic programming language. ***Probabilistic programming*** [25] has recently emerged as a powerful tool in Bayesian inference. Probabilistic programming systems allow users to specify complex statistical models as programs in a precise yet intuitive way, and automate the Bayesian inference task. This allows practitioners to focus on modelling, leaving the development of general-purpose inference algorithms to experts. Consequently, probabilistic programming systems such as Stan [3] enjoy increasing popularity among statisticians and data scientists. We describe a programming language, called SGCL (statistical guarded command language), that extends pGCL (probabilistic GCL) [17]. This language is carefully designed to be simple yet expressive, and just restrictive enough to enable exact Bayesian inference on all programs that can be written in it.

**Contributions**     We provide a new framework for exact inference on discrete Bayesian models.

(a) Our method is applicable to a large class of *discrete models with infinite support*, in particular time series models of count data, such as autoregressive models for population dynamics, Bayesian switchpoint models, mixture models, and hidden Markov models. To our knowledge, no exact inference method for all but the simplest population models was known before.
(b) The models are specified in a *probabilistic programming language (PPL)*, which provides flexibility in model definition, thus facilitating model construction. Our PPL supports stochastic branching, continuous and discrete priors, discrete observations, and conditioning on events involving discrete variables.
(c) Every program written in the language can be *translated to a generating function* that represents the posterior distribution in an automatic and provably correct way. From this generating function, one can extract posterior *mean, variance, and higher moments*, as well as the posterior *probability masses* (for a discrete distribution).
(d) We have built an *optimized tool*, called Genfer ("GENerating Functions for inFERence"), that takes a probabilistic program as input and *automatically computes* the aforementioned set of descriptive statistics about the posterior distribution.
(e) We demonstrate that (1) on benchmarks with finite support, Genfer's performance is often better than existing *exact inference tools*, and (2) on a range of real-world examples that no existing exact tool supports, Genfer is competitive with *approximate Monte Carlo methods*, while achieving zero approximation error.

**Related Work**     Computing the exact posterior distribution of probabilistic programs is intractable in general as it requires analytical solutions to integrals [9]. For this reason, existing systems either restrict the programming language to allow only tractable constructs (this is our approach) or cannot guarantee successful inference. In the former category are Dice [16], which only supports finite discrete distributions, and SPPL [22], which supports some infinite-support distributions but requires finite discrete priors. Both are based on probabilistic circuits [5] or extensions thereof, which allow for efficient and exact inference. In the latter category are the systems PSI [9] and Hakaru [21], which do not impose syntactic restrictions. They rely on computer algebra techniques to find a closed-form solution for the posterior. Such a form need not exist in general and, even if it does, the system may fail to find it and the running time is unpredictable and unscalable [22]. None of the case studies featured in our evaluation (Section 5.2) can be handled by the aforementioned tools.

Probability generating functions are a useful tool in probability theory to study random variables with infinite support, e.g. in the context of branching processes [7, Chapter 12]. In the context of Bayesian inference, probabilistic generating circuits (PGCs) leverage them to boost the expressiveness of probabilistic circuits [28] and enable efficient inference [15]. However, PGCs cannot handle random variables with infinite support, which is the focus of our approach. Generating functions have also been applied to probabilistic graphical models: Winner and Sheldon [26] find a symbolic representation of the generating function for a Poisson autoregressive model and extract posterior probabilities from it. Subsequently, Winner et al. [27] extend this model to latent variable distributions other than the Poisson distribution, where symbolic manipulation is no longer tractable. Instead they evaluate generating functions and their derivatives using automatic differentiation, which enables exact inference for graphical models. Probabilistic programming is an elegant way of generalizing graphical models, allowing a much richer representation of models [10, 25, 12]. Our contribution here is a new framework for exact inference on Bayesian models via probabilistic programming.

In the context of discrete probabilistic programs without conditioning, Klinkenberg et al. [17] use generating functions to (manually) analyze loop invariants and determine termination probabilities. In follow-up work, Chen et al. [4] extend these techniques to automatically check (under certain restrictions) whether a looping program generates a specified distribution. Their analysis tool Prodigy recently gained the ability to perform Bayesian inference as well [18]. It supports discrete distributions with infinite support (but no continuous priors) and is less scalable than our automatic-differentiation approach (see Section 5) since it relies on computer algebra.

**Limitations**    Exact posterior inference is already PSPACE-hard for probabilistic programs involving only finite discrete distributions [16, Section 6]. It follows that our method cannot always be performant and has to restrict the supported class of probabilistic programs. Indeed, our programming language forbids certain constructs, such as nonlinear transformations and observations of continuous variables, in order to preserve a closed-form representation of the generating functions (Section 2). For the same reason, our method cannot compute probability density functions for continuous parameters (but probability masses for discrete parameters and exact moments for all parameters are fine). Regarding performance, the running time of inference is polynomial in the numbers observed in the program and exponential in the number of program variables (Section 4). Despite these limitations, our approach shows that exact inference is possible in the first place, and our evaluation (Section 5) demonstrates support for many real-world models and efficient exact inference in practice.

## 2    Bayesian Probabilistic Programming

Probabilistic programming languages extend ordinary programming languages with two additional constructs: one for **_sampling_** from probability distributions and one for **_conditioning_** on observed values. We first discuss a simple program that can be written in our language using a simplified example based on population ecology (cf. [26]).

Suppose you're a biologist trying to estimate the size of an animal population migrating into a new habitat. The immigration of animals is often modeled using a Poisson distribution. You cannot count the animals exhaustively (otherwise we wouldn't need estimation techniques), so we assume that each individual is observed independently with a certain probability; in other words, the count is binomially distributed. For simplicity, we assume that the rate of the Poisson and the probability of the binomial distribution are known, say 20 and 0.1. (For more realistic population models, see Section 5.) As a generative model, this would be written as $X \sim \mathsf{Poisson}(20); Y \sim \mathsf{Binomial}(X, 0.1)$.

Suppose you observe $Y = 2$ animals. The Bayesian inference problem is to compute the **_posterior distribution_** $\mathbb{P}(X = x \mid Y = 2)$ given by Bayes' rule as $\mathbb{P}(X = x \mid Y = 2) = \frac{\mathbb{P}(X=x)\,\mathbb{P}(Y=2|X=x)}{\mathbb{P}(Y=2)}$, where $\mathbb{P}(X = x)$ is called the **_prior probability_**, $\mathbb{P}(Y = 2 \mid X = x)$ the **_likelihood_** and $\mathbb{P}(Y = 2)$ the **_evidence_** or **_normalization constant_**.

**Example 2.1.** In our probabilistic programming language, this simplified population model would be expressed as:

$$X \sim \mathsf{Poisson}(20); Y \sim \mathsf{Binomial}(X, 0.1); \mathsf{observe}\, Y = 2;$$

The syntax looks similar to generative model notation except that the observations are expressible as a command. Such a program denotes a joint probability distribution of its program variables, which are viewed as random variables. The program statements manipulate this joint distribution. After the first two sampling statements, the distribution has the probability mass function (PMF) $p_1(x, y) = \mathbb{P}[X = x, Y = y] = \mathsf{Poisson}(x; 20) \cdot \mathsf{Binomial}(y; x, 0.1)$. Observing $Y = 2$ restricts this to the PMF $p_2(x, y) = \mathbb{P}[X = x, Y = y = 2]$, which equals $\mathbb{P}[X = x, Y = 2] = p_1(x, 2)$ if $y = 2$ and 0 otherwise. Note that this is not a probability, but a **_subprobability_**, distribution because the total mass is less than 1. So, as a final step, we need to **_normalize_**, i.e. rescale the subprobability distribution back to a probability distribution. This corresponds to the division by the evidence in Bayes' rule, yielding the PMF $p_3(x, y) = \frac{p_2(x,y)}{\sum_{x,y} p_2(x,y)}$. To obtain the posterior $\mathbb{P}[X = x \mid Y = 2]$, which is a distribution of the single variable $X$, not the joint of $X$ and $Y$, we need to marginalize $p_3$ and find $\sum_y p_3(x, y) = \frac{\sum_y p_2(x,y)}{\sum_{x,y} p_2(x,y)} = \frac{p_1(x,2)}{\sum_x p_1(x,2)} = \frac{\mathbb{P}[X=x,Y=2]}{\mathbb{P}[Y=2]} = \mathbb{P}[X = x \mid Y = 2]$, as desired.

**Programming constructs**    Next we describe our probabilistic programming language more formally. It is based on the **_probabilistic guarded command language (pGCL)_** from [17] but augments

it with statistical features like conditioning on events and normalization, which is why we call it ***Statistical GCL (SGCL)***. Each program operates on a fixed set of variables $\boldsymbol{X} = \{X_1, \ldots, X_n\}$ taking values in $\mathbb{R}_{\geq 0}$. A program consists of a list of statements $P_1; P_2; \ldots; P_m$. The simplest statement is skip, which does nothing and is useful in conditionals (like pass in Python). Variables can be transformed using affine maps, e.g. $X_2 := 2X_1 + 7X_3 + 2$ (note that the coefficients must be non-negative to preserve nonnegativity of the variables). Programs can branch on the value of a (discrete) variable (e.g. if $X_k \in \{1, 3, 7\} \{\ldots\}$ else $\{\ldots\}$), and sample new values for variables from distributions (e.g. $X_k \sim \mathsf{Binomial}(10, 0.5)$ or $X_k \sim \mathsf{Binomial}(X_j, 0.5)$). The supported distributions are Bernoulli, Categorical, Binomial, Uniform (both discrete and continuous), NegBinomial, Geometric, Poisson, Exponential, and Gamma. They need to have constant parameters except for the ***compound distributions*** $\mathsf{Binomial}(X, p)$, $\mathsf{NegBinomial}(X, p)$, $\mathsf{Poisson}(\lambda \cdot X)$, and $\mathsf{Bernoulli}(X)$, where $X$ can be a variable. One can observe events involving discrete variables (e.g. observe $X_k \in \{3, 5\}$) and values from (discrete) distributions directly (e.g. observe $3 \sim \mathsf{Binomial}(X_j, 0.5)$). Note that observe $m \sim D$ can be seen as a convenient abbreviation of $Y \sim D$; observe $Y = m$ with a fresh variable $Y$. After an observation, the variable distribution is usually not a probability distribution anymore, but a ***subprobability distribution*** ("the numerator of Bayes' rule").

**Syntax**    In summary, the syntax of programs $P$ has the following BNF grammar:

$$P ::= \mathsf{skip} \mid P_1; P_2 \mid X_k := a_1 X_1 + \cdots + a_n X_n + c \mid \mathsf{if}\ X_k \in A\ \{P_1\}\ \mathsf{else}\ \{P_2\}$$
$$\mid X_k \sim D \mid X_k \sim D(X_j) \mid \mathsf{observe}\ X_k \in A \mid \mathsf{observe}\ m \sim D \mid \mathsf{observe}\ m \sim D(X_j)$$

where $P$, $P_1$ and $P_2$ are subprograms; $a_1, \ldots, a_n, c \in \mathbb{R}_{\geq 0}$; $m \in \mathbb{N}$; $A$ is a finite subset of the naturals; and $D$ is a supported distribution. To reduce the need for temporary variables, the abbreviations $+\sim$, $+=$, and if $m \sim D \{\ldots\}$ else $\{\ldots\}$ are available (see Section 4). A fully formal description of the language constructs can be found in Appendix A.

**Restrictions**    Our language imposes several syntactic restrictions to guarantee that the generating function of any program admits a closed form, which enables exact inference (see Section 3.1): (a) only affine functions are supported (e.g. no $X^2$ or $\exp(X)$), (b) only comparisons between variables and constants are supported (e.g. no test for equality $X = Y$), (c) only observations from discrete distributions on $\mathbb{N}$ and only comparisons of such random variables are supported (e.g. no observe $1.5 \sim \mathsf{Normal}(0, 1)$), (d) a particular choice of distributions and their composites is supported, (e) loops or recursion are not supported. A discussion of possible language extensions and relaxations of these restrictions can be found in Appendix B.3.

## 3    Generating Functions

Consider Example 2.1. Even though the example is an elementary exercise in probability, it is challenging to compute the normalizing constant, because one needs to evaluate an infinite sum: $\mathbb{P}[Y = 2] = \sum_{x \in \mathbb{N}} \mathbb{P}[Y = 2 \mid X = x]\, \mathbb{P}[X = x] = \sum_{x \in \mathbb{N}} \binom{x}{2} 0.1^2\, 0.9^{x-2} \cdot \frac{20^x e^{-20}}{x!}$. It turns out to be $2e^{-2}$ (see Example 3.1), but it is unclear how to arrive at this result in an automated way. If $X$ had been a continuous variable, we would even have to evaluate an integral. We will present a technique to compute such posteriors mechanically. It relies on probability generating functions, whose definition includes an infinite sum or integral, but which often admit a closed form, thus enabling the exact computation of posteriors.

**Definition**    Probability generating functions are a well-known tool in probability theory to study random variables and their distributions, especially discrete ones. The probability generating function of a random variable $X$ is defined to be the function $G(x) := \mathbb{E}[x^X]$. For a discrete random variable supported on $\mathbb{N}$, this can also be written as a ***power series*** $G(x) = \sum_{n \in \mathbb{N}} \mathbb{P}[X = n] \cdot x^n$. Since we often deal with subprobability distributions, we omit "probability" from the name and refer to $G(x) := \mathbb{E}_{X \sim \mu}[x^X]$, where $\mu$ is a subprobability measure, simply as a ***generating function*** (GF). For continuous variables, it is often called ***factorial moment generating function***, but we stick with the former name in all contexts. We will use the notation $\mathsf{gf}(\mu)$ or $\mathsf{gf}(X)$ for the GF of $\mu$ or $X$. Note that for discrete random variables supported on $\mathbb{N}$, the GF is always defined on $[-1, 1]$, and for continuous ones at $x = 1$, but it need not be defined at other $x$.

**Probability masses and moments**    Many common distributions admit a ***closed form*** for their GF (see Table 1). In such cases, GFs are a compact representation of a distribution, even if it has infinite or continuous support like the Poisson or Exponential distributions, respectively. Crucially, one can

Table 1: GFs for common distributions with constant (left) and random variable parameters (right)

| Distribution $D$ | $\mathsf{gf}(X \sim D)(x)$ | Distribution $D(Y)$ | $\mathsf{gf}(X \sim D(Y))(x)$ |
|---|---|---|---|
| Binomial$(n, p)$ | $(px + 1 - p)^n$ | Binomial$(Y, p)$ | $\mathsf{gf}(Y)(1 - p + px)$ |
| Geometric$(p)$ | $\frac{p}{1-(1-p)x}$ | NegBinomial$(Y, p)$ | $\mathsf{gf}(Y)\left(\frac{p}{1-(1-p)x}\right)$ |
| Poisson$(\lambda)$ | $e^{\lambda(x-1)}$ | Poisson$(\lambda \cdot Y)$ | $\mathsf{gf}(Y)(e^{\lambda(x-1)})$ |
| Exponential$(\lambda)$ | $\frac{\lambda}{\lambda-\log x}$ | Bernoulli$(Y)$ | $1 + (x - 1) \cdot (\mathsf{gf}(Y))'(1)$ |

extract **_probability masses_** and **_moments_** of a distribution from its generating function. For discrete random variables $X$, $P[X = n]$ is the $n$-th coefficient in the power series representation of $G$, so can be computed as the **_Taylor coefficient_** at 0: $\frac{1}{n!}G^{(n)}(0)$ (hence the name "*probability* generating function"). For a discrete or continuous random variable $X$, its expected value is $\mathbb{E}[X] = G'(1)$, and more generally, its $n$-th **_factorial moment_** is $\mathbb{E}[X(X - 1)\cdots(X - n + 1)] = \mathbb{E}[\frac{\mathrm{d}^n}{\mathrm{d}x^n}x^X]|_{x=1} = G^{(n)}(1)$ (hence the name "*factorial moment* generating function"). The raw and central moments can easily be computed from the factorial moments. For instance, the variance is $\mathbb{V}[X] = G''(1)+G'(1)-G'(1)^2$. We will exploit these properties of generating functions through automatic differentiation.

**Multivariate case** The definition of GFs is extended to **_multidimensional distributions_** in a straightforward way: for random variables $\boldsymbol{X} = (X_1, \ldots, X_n)$, their GF is the function $G(\boldsymbol{x}) := \mathbb{E}[\boldsymbol{x}^{\boldsymbol{X}}]$ where we write $\boldsymbol{x} := (x_1, \ldots, x_n)$ and $\boldsymbol{x}^{\boldsymbol{X}} := x_1^{X_1}\cdots x_n^{X_n}$. We generally follow the convention of using uppercase letters for random and program variables, and lowercase letters for the corresponding parameters of the generating function. **_Marginalization_** can also be expressed in terms of generating functions: to obtain the GF $\tilde{G}$ of the joint distribution of $(X_1, \ldots, X_{n-1})$, i.e. to marginalize out $X_n$, one simply substitutes 1 for $x_n$ in $G$: $\tilde{G}(x_1, \ldots, x_{n-1}) = G(x_1, \ldots, x_{n-1}, 1)$. This allows us to compute probability masses and moments of a random variable in a joint distribution: we marginalize out all the other variables and then use the previous properties of the derivatives.

### 3.1 Translating programs to generating functions

The standard way of describing the meaning of probabilistic programs is assigning (sub-)probability distributions to them. An influential example is Kozen's **_distribution transformer_** semantics [19], where each program statement transforms the **_joint distribution_** of all the variables $\boldsymbol{X} = (X_1, \ldots, X_n)$. Since Kozen's language does not include observations, we present the full semantics in Appendix A. We call this the **_standard semantics_** of a probabilistic program and write $[\![P]\!]_{\mathsf{std}}(\mu)$ for the transformation of $\mu$ by the program $P$. As a last step, the subprobability distribution $\mu$ has to be **_normalized_**, which we write $\mathsf{normalize}(\mu) := \frac{\mu}{\int \mathrm{d}\mu}$. This reduces the Bayesian inference problem for a given program to computing its semantics, starting from the joint distribution $\mathsf{Dirac}(\boldsymbol{0}_n)$ in which all $n$ variables are initialized to 0 with probability 1. While mathematically useful, the distribution transformer semantics is hardly amenable to computation as it involves integrals and infinite sums.

Instead, we shall compute the *generating function* of the posterior distribution represented by the probabilistic program. Then we can extract posterior probability masses and moments using automatic differentiation. Each statement in the programming language transforms the generating function of the distribution of program states, i.e. the joint distribution of the values of the variables $\boldsymbol{X} = (X_1, \ldots, X_n)$. Initially, we start with the constant function $G = \boldsymbol{1}$, which corresponds to all variables being initialized with 0 since $\mathbb{E}[x^0] = 1$.

The **_generating function semantics_** of a program $[\![P]\!]_{\mathsf{gf}}$ describes how to transform $G$ to the generating function $[\![P]\!]_{\mathsf{gf}}(G)$ for the distribution at the end. It is defined in Table 2, where the update notation $\boldsymbol{x}[i \mapsto a]$ denotes $(x_1, \ldots, x_{i-1}, a, x_{i+1}, \ldots, x_n)$ and $\boldsymbol{1}_n$ means the $n$-tuple $(1, \ldots, 1)$. The first five rules were already described in [4], so we only explain them briefly: skip leaves everything unchanged and $P_1; P_2$ chains two statements by transforming with $[\![P_1]\!]_{\mathsf{gf}}$ and then $[\![P_2]\!]_{\mathsf{gf}}$. To explain linear assignments, consider the case of only two variables: $X_1 := 2X_1 + 3X_2 + 5$. Then

$$[\![P]\!]_{\mathsf{gf}}(G)(\boldsymbol{x}) = \mathbb{E}[x_1^{2X_1+3X_2+5}x_2^{X_2}] = x_1^5 \mathbb{E}[(x_1^2)^{X_1}(x_1^3 x_2)^{X_2}] = x_1^5 \cdot G(x_1^2, x_1^3 x_2).$$

For conditionals if $X_k \in A \{P_1\}$ else $\{P_2\}$, we split the generating function $G$ into two parts: one where the condition is satisfied ($G_{X_k \in A}$) and its complement ($G - G_{X_k \in A}$). The former is transformed

Table 2: Generating function semantics of programming constructs

| Language construct $P$ | $[\![P]\!]_{\mathsf{gf}}(G)(\boldsymbol{x})$ |
|---|---|
| skip | $G(\boldsymbol{x})$ |
| $P_1 ; P_2$ | $[\![P_2]\!]_{\mathsf{gf}}([\![P_1]\!]_{\mathsf{gf}}(G))(\boldsymbol{x})$ |
| $X_k := \mathbf{a}^\top \boldsymbol{X} + c$ | $x_k^c \cdot G(\boldsymbol{x}')$ where $x_k' := x_k^{a_k}$ and $x_i' := x_i x_k^{a_i}$ for $i \neq k$ |
| if $X_k \in A \{P_1\}$ else $\{P_2\}$ | $[\![P_1]\!]_{\mathsf{gf}}(G_{X_k \in A}) + [\![P_2]\!]_{\mathsf{gf}}(G - G_{X_k \in A})$ |
| | where $G_{X_k \in A}(\mathbf{x}) = \sum_{i \in A} \frac{\partial_k^i G(\mathbf{x}[k \mapsto 0])}{i!} x_k^i$ |
| $X_k \sim D$ | $G(\mathbf{x}[k \mapsto 1]) \cdot \mathsf{gf}(D)(x_k)$ |
| $X_k \sim D(X_j)$ | $G(\mathbf{x}[k \mapsto 1, j \mapsto x_j \cdot \mathsf{gf}(D(1))(x_k)])$ |
| | for $D \in \{\, \mathsf{Binomial}(-, p),\ \mathsf{NegBinomial}(-, p),\ \mathsf{Poisson}(\lambda \cdot -) \,\}$ |
| | $G(\boldsymbol{x}[k \mapsto 1]) + x_j(x_k - 1) \cdot \partial_j G(\boldsymbol{x}[k \mapsto 1])$ for $D = \mathsf{Bernoulli}(-)$ |
| observe $X_k \in A$ | $G_{X_k \in A}(\mathbf{x}) = \sum_{i \in A} \frac{\partial_k^i G(\mathbf{x}[k \mapsto 0])}{i!} x_k^i$ |
| Normalization | $\mathsf{normalize}(G) := \frac{G}{G(\mathbf{1}_n)}$ |

by the then-branch $[\![P_1]\!]_{\mathsf{gf}}$, the latter by the else-branch $[\![P_2]\!]_{\mathsf{gf}}$. The computation of $G_{X_k \in A}$ is best understood by thinking of $G$ as a power series where we keep only the terms where the exponent of $x_k$ is in $A$. Sampling $X_k \sim D$ from a distribution with constant parameters works by first marginalizing out $X_k$ and then multiplying by the generating function of $D$ with parameter $x_k$.

The first new construct is sampling $X_k \sim D(X_j)$ from compound distributions (see Appendix B for a detailed explanation). Observing events $X_k \in A$ uses $G_{X_k \in A}$ like in conditionals, as explained above. Just like the subprobability distribution defined by a program has to be normalized as a last step, we have to normalize the generating function. The normalizing constant is calculated by marginalizing out all variables: $G(1, \ldots, 1)$. So we obtain the generating function representing the normalized posterior distribution by rescaling with the inverse: $\mathsf{normalize}(G) := \frac{G}{G(\mathbf{1}_n)}$. These intuitions can be made rigorous in the form of the following theorem, which is proven in Appendix B.2.

**Theorem 3.1.** *The GF semantics is correct w.r.t. the standard semantics: for any SGCL program $P$ and subprobability distribution $\mu$ on $\mathbb{R}_{\geq 0}^n$, we have $[\![P]\!]_{\mathsf{gf}}(\mathsf{gf}(\mu)) = \mathsf{gf}([\![P]\!]_{\mathsf{std}}(\mu))$. In particular, it correctly computes the GF of the posterior distribution of $P$ as $\mathsf{normalize}([\![P]\!]_{\mathsf{gf}}(\mathbf{1}))$. Furthermore, there is some $R > 1$ such that $[\![P]\!]_{\mathsf{gf}}(\mathbf{1})$ and $\mathsf{normalize}([\![P]\!]_{\mathsf{gf}}(\mathbf{1}))$ are defined on $\{\boldsymbol{x} \in \mathbb{R}^n \mid Q_i < x_i < R\}$ where $Q_i = -R$ if the variable $X_i$ is supported on $\mathbb{N}$ and $Q_i = 0$ otherwise.*

**Novelty** The semantics builds upon [17, 4]. To our knowledge, the GF semantics of the compound distributions $\mathsf{Poisson}(\lambda \cdot X_j)$ and $\mathsf{Bernoulli}(X_j)$ is novel, and the former is required to support most models in Section 5. While the GF of observations has been considered in the context of a specific model [26], this has not been done in the general context of a probabilistic programming language before. More generally, previous works involving GFs only considered discrete distributions, whereas we also allow sampling from continuous distributions. This is a major generalization and requires different proof techniques because the power series representation $\sum_{i \in \mathbb{N}} \mathbb{P}[X = i] x^i$, on which the proofs in [26, 17, 4] rely, is not valid for continuous distributions.

**Example 3.1** (GF translation). Consider Example 2.1. We can find the posterior distribution mechanically by applying the rules from the GF semantics. We start with the GF $A(x, y) = \mathbb{E}[x^0 y^0] = 1$ corresponding to $X$ and $Y$ being initialized to 0. Sampling $X$ changes this to GF $B(x, y) = A(1, y)e^{20(x-1)} = e^{20(x-1)}$. Sampling $Y$ yields $C(x, y) = B(x(0.1y + 0.9), 1) = e^{2x(y+9)-20}$. Observing $Y = 2$ yields $D(x, y) = \frac{1}{2!} y^2 \frac{\partial^2}{\partial y^2} C(x, 0) = 2x^2 y^2 e^{18x-20}$. To normalize, we divide by $D(1, 1) = 2e^{-2}$, obtaining $E(x, y) = \frac{D(x,y)}{D(1,1)} = x^2 y^2 e^{18(x-1)}$ since $A(x, y) = 1$.

As described above, we can extract from this GF the posterior probability of, for example, exactly 10 individuals $\mathbb{P}[X = 10] = \frac{1}{10!} \frac{\partial^{10}}{\partial x^{10}} E(0, 1) = 991796451840 e^{-18}$ and the expected value of the posterior $\mathbb{E}[X] = \frac{\partial}{\partial x} E(1, 1) = 20$.

## 4 Implementation & Optimizations

The main difficulty in implementing the GF semantics is the computation of the partial derivatives. A natural approach (as followed by [4, 18]) is to manipulate symbolic representations of the generating

functions and to use computer algebra for the derivatives. However this usually scales badly, as demonstrated by Winner et al. [27], because the size of the generating functions usually grows quickly with the data conditioned on. To see why, note that every observe $X_k = d$ statement in the program is translated to a $d$-th partial derivative. Since probabilistic programs tend to contain many data points, it is common for the total order of derivatives to be in the hundreds. The size of the symbolic representation of a function can (and typically does) grow exponentially in the order of the derivative: the derivative of the product of two functions $f \cdot g$ is the sum of two products $f' \cdot g + f \cdot g'$, so the representation doubles in size. Hence the running time would be $\Omega(2^d)$ where $d$ is the sum of all observed values, which is clearly unacceptable.

Instead, we exploit the fact that we do not need to generate the full representation of a GF, but merely to *evaluate it and its derivatives*. We implement our own ***automatic differentiation*** framework for this because existing ones are not designed for computing derivatives of order greater than, say, 4 or 5. In fact, it is more efficient to work with Taylor polynomials instead of higher derivatives directly. Winner et al. [27] already do this for the population model (with only one variable), and we extend this to our more general setting with multiple variables, requiring ***multivariate Taylor polynomials***. In this approach, derivatives are the easy part as they can be read off the Taylor coefficients, but the composition of Taylor polynomials is the bottleneck. Winner et al. [27] use a naive $O(d^3)$ approach, which is fast enough for their single-variable use case.

**Running time** For $n$ variables, naive composition of Taylor polynomials takes $O(d^{3n})$ time, where $d$ is the sum of all observations in the program, i.e. the total order of differentiation, i.e. the degree of the polynomial. Note that this is polynomial in $d$, contrary to the symbolic approach, but exponential in the number of variables $n$. This is not as bad as it seems because in many cases, the number of variables can be kept to one or two, as opposed to the values of data points (such as the models from Section 5). In fact, we exploit the specific composition structure of generating functions to achieve $O(d^3)$ for $n = 1$ and $O(d^{n+3})$ for $n \geq 2$ in the worst case, while often being faster in practice. Overall, our implementation takes $O(sd^{n+3})$ time in the worst case, where $s$ is the number of statements in the program, $d$ is the sum of all observed values, and $n$ is the number of program variables (see Appendix C.3).

**Reducing the number of variables** Given the exponential running time in $n$, it is crucial to reduce the number of variables when writing probabilistic programs. For one thing, program variables that are no longer needed can often be reused for a different purpose later. Furthermore, assignment and sampling can be combined with addition: $X_k += \ldots$ stands for $X_{n+1} := \ldots; X_k := X_k + X_{n+1}$ and $X_k += \sim D$ for $X_{n+1} \sim D; X_k := X_k + X_{n+1}$. The GFs for these statements can easily be computed without introducing the temporary variable $X_{n+1}$. Similarly, in observe statements and if-conditions, we use the shorthand $m \sim D$ with $m \in \mathbb{N}$ for the event $X_{n+1} = m$ where $X_{n+1} \sim D$. Probabilistic programs typically contain many such observations, in particular from compound distributions. Hence it is worthwhile to optimize the generating function to avoid this extra variable $X_{n+1}$. Winner and Sheldon [26] can avoid an extra variable for a compound binomial distribution in the context of a specific model. We extend this to our more general setting with continuous variables, and also present optimized semantics for compound Poisson, negative binomial, and Bernoulli distributions. In fact, this optimization is essential to achieving good performance for many of the examples in Section 5. The optimized translation and its correctness proof can be found in Appendix C.2.

**Implementation** Our tool ***Genfer*** reads a program file and outputs the ***posterior mean***, ***variance***, ***skewness***, and ***kurtosis*** of a specified variable. For discrete variables supported on $\mathbb{N}$, it also computes the ***posterior probability masses*** up to a configurable threshold. To have tight control over performance, especially for the multivariate Taylor polynomials, Genfer is written in Rust [20], a safe systems programming language. Our implementation is available on GitHub: github.com/fzaiser/genfer

**Numerical issues** Genfer can use several number formats for its computations: 64-bit floating point (the default and fastest), floating point with a user-specified precision, and rational numbers (if no irrational numbers occur). To ensure that the floating-point results are numerically stable, we also implemented interval arithmetic to bound the rounding errors. Initially, we found that programs with continuous distributions led to catastrophic cancellation errors, due to the logarithmic term in their GFs, whose Taylor expansion is badly behaved. We fixed this problem with a slightly modified representation of the GFs, avoiding the logarithms (details in Appendix C). For all the examples in Section 5.2, our results are accurate up to at least 5 significant digits.

Table 3: Comparison of inference times of tools for exact inference on PSI's benchmarks [9]

| Tool | Genfer (FP) | Dice (FP) | Genfer ($\mathbb{Q}$) | Dice ($\mathbb{Q}$) | Prodigy | PSI |
|---|---|---|---|---|---|---|
| alarm (F) | **0.0005s** | 0.0067s | **0.0012s** | 0.0066s | 0.011s | 0.0053s |
| clickGraph (C) | **0.11s** | unsupported | **3.4s** | unsupported | unsupported | 46s |
| clinicalTrial (C) | **150s** | unsupported | **1117s** | unsupported | unsupported | timeout |
| clinicalTrial2 (C) | **0.0024s** | unsupported | **0.031s** | unsupported | unsupported | 0.46s |
| digitRecognition (F) | **0.021s** | 0.83s | **0.11s** | 2.7s | 31s | 146s |
| evidence1 (F) | **0.0002s** | 0.0057s | **0.0003s** | 0.0056s | 0.0030s | 0.0016s |
| evidence2 (F) | **0.0002s** | 0.0056s | **0.0004s** | 0.0057s | 0.0032s | 0.0018s |
| grass (F) | **0.0008s** | 0.0067s | **0.0044s** | 0.0067s | 0.019s | 0.014s |
| murderMystery (F) | **0.0002s** | 0.0055s | **0.0003s** | 0.0057s | 0.0028s | 0.0021s |
| noisyOr (F) | **0.0016s** | 0.0085s | 0.019s | **0.0088s** | 0.21s | 0.055s |
| twoCoins (F) | **0.0002s** | 0.0054s | **0.0003s** | 0.0057s | 0.0032s | 0.0017s |

## 5 Empirical Evaluation

### 5.1 Comparison with exact inference methods

We compare our tool Genfer with the following tools for exact Bayesian inference: Dice [16], which uses weighted model counting, PSI [9], which manipulates density functions using computer algebra, and Prodigy [18], which is based on generating functions like Genfer, but uses computer algebra instead of automatic differentiation.[1] We evaluate them on the PSI benchmarks [9], excluding those that only PSI supports, e.g. due to observations from continuous distributions. Most benchmarks only use finite discrete distributions (labeled "F"), but three feature continuous priors (labeled "C").

We measured the wall-clock inference time for each tool, excluding startup time and input file parsing, and recorded the minimum from 5 consecutive runs with a one-hour timeout.[2] Dice and Genfer default to floating-point (FP) numbers, whereas Prodigy and PSI use rational numbers, which is slower but prevents rounding errors. For a fair comparison, we evaluated all tools in rational mode and separately compared Dice with Genfer in FP mode. The results (Table 3) demonstrate Genfer's speed even on finite discrete models, despite our primary focus on models with infinite support.

### 5.2 Comparison with approximate inference methods

It is impossible to compare our approach with other exact inference methods on realistic models with infinite support (Section 5.3): the scalable systems Dice [16] and SPPL [22] don't support such priors and the symbolic solvers PSI [9] and Prodigy [18] run out of memory or time out after an hour.

**Truncation** As an alternative, we considered approximating the posterior by truncating discrete distributions with infinite support. This reduces the problem to finite discrete inference, which is more amenable to exact techniques. We decided against this approach because Winner and Sheldon [26] already demonstrated its inferiority to GF-based exact inference on their graphical model. Moreover, it is harder to truncate general probabilistic programs, and even impossible for continuous priors.

**Monte-Carlo inference** Hence, we compare our approach with Monte Carlo inference methods. Specifically, we choose the Anglican [24] probabilistic programming system because it offers the best built-in support for discrete models with many state-of-the-art inference algorithms. Other popular systems are less suitable: Gen [6] specializes in programmable inference; Stan [3], Turing [8], and Pyro [1] mainly target continuous models; and WebPPL's [11] discrete inference algorithms are less extensive than Anglican's (e.g. no support for interacting particle MCMC).

**Methodology** The approximation error of a Monte Carlo inference algorithm depends on its *settings*[3] (e.g. the number of particles for SMC) and decreases with the number of samples the longer it is run. To ensure a fair comparison, we use the following setup: for each inference problem, we run several inference algorithms with various *configurations* (settings and sampling budgets) and

---

[1]We were not able to run Hakaru [21], which uses the computer algebra system Maple as a backend, but we do not expect it to produce better results than the other systems, given that PSI is generally more performant [9].

[2]PSI was run with the experimental flag `--dp` on finite discrete benchmarks for improved performance.

[3]See `https://probprog.github.io/anglican/inference/` for a full list of Anglican's options.

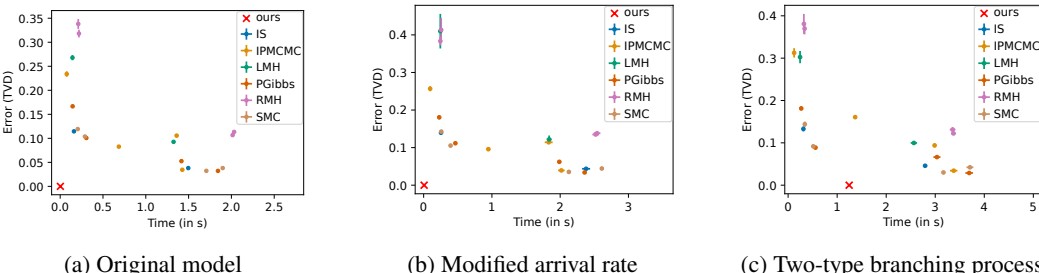

| (a) Original model | (b) Modified arrival rate | (c) Two-type branching process |

Figure 1: Comparison of the population model and its modifications with approximate inference: mean and standard error of the computation time and TVD over 20 repeated runs.

report the approximation error and the elapsed time. To measure the quality of the approximation, we report the ***total variation distance (TVD)*** between the exact solution and the approximate posterior distribution, as well as the ***approximation errors*** of the posterior mean $\mu$, standard deviation $\sigma$, skewness (third standardized moment) $S$, and kurtosis $K$. To ensure invariance of our error metrics under translation and scaling of the distribution, we compute the error of the mean as $\frac{|\hat{\mu}-\mu|}{\sigma}$, where $\mu$ is the true posterior mean, $\hat{\mu}$ its approximation, and $\sigma$ the true posterior standard deviation; the relative error of the standard deviation $\sigma$ (because $\sigma$ is invariant under translation but not scaling), and the absolute error of skewness and kurtosis (because they are invariant under both translation and scaling). To reduce noise, we average all these error measures and the computation times over 20 runs and report the standard error as error bars. We run several well-known ***inference algorithms*** implemented in Anglican: importance sampling (IS), Lightweight Metropolis-Hastings (LMH), Random Walk Metropolis-Hastings (RMH), Sequential Monte Carlo (SMC), Particle Gibbs (PGibbs), and interacting particle MCMC (IPMCMC). For comparability, in all experiments, each algorithm is run with two sampling budgets and, if possible, two different settings (one being the defaults) for a total of four configurations. The ***sampling budgets*** were 1000 or 10000, because significantly lower sample sizes gave unusable results and significantly higher sample sizes took much more time than our exact method. We discard the first 20% of the samples, a standard procedure called "burn-in".

Note that this setup is generous to the approximate methods because we only report the average time for one run of each configuration. However, in practice, one does not know the best configuration, so the algorithms need to be run several times with different settings. By contrast, our method requires only one run because the result is exact.

### 5.3 Benchmarks with infinite-support distributions

**Population ecology**     Our first benchmark comes from [26, 27] and models animal populations. We have seen a simplified version in Example 2.1. Here we model a population $N_k$ at time steps $k = 0, \ldots, m$. At each time step, there is a Poisson-distributed number of new arrivals, which are added to the binomially distributed number of survivors from the previous time step. Each individual is observed with a fixed probability $\delta$, so the number of observed individuals is binomially distributed:

$$New_k \sim \mathsf{Poisson}(\lambda_k); \quad Survivors_k \sim \mathsf{Binomial}(N_{k-1}, \delta);$$
$$N_k := New_k + Survivors_k; \quad \text{observe } y_k \sim \mathsf{Binomial}(N_k, \rho);$$

where the model parameters $\lambda_k \in \mathbb{R}, \delta \in [0, 1]$ are taken from [26]; the detection probability $\rho$ is set to 0.2 ([26] considers a range of values, but we pick one for space reasons); and the observed number $y_k \in \mathbb{N}$ of individuals in the population at time step $k$ is simulated from the same ground truth as [26]. The goal is to infer the final number $N_m$ of individuals in the population. We set the population size model parameter and hence the observed values which influence the running time to be 4 times larger than the largest in [26] (see details in Appendix D.3). The results (Fig. 1a) show that our method is superior to MCMC methods in both computation time and accuracy since it is exact.

**Modifications**     While this model was already solved exactly in [26], our probabilistic programming approach makes it trivial to modify the model, since the whole inference is automated and one only needs to change a few lines of the program: (a) we can model the possibility of natural disasters affecting the offspring rate with a conditional: $Disaster \sim \mathsf{Bernoulli}(0.1)$; if $Disaster = 1 \{New_k \sim \mathsf{Poisson}(\lambda')\}$ else $\{New_k \sim \mathsf{Poisson}(\lambda)\}$, or (b) instead of a single population, we can model populations of two kinds of individuals that interact, i.e. a *multitype* branching process (see Fig. 1c). None

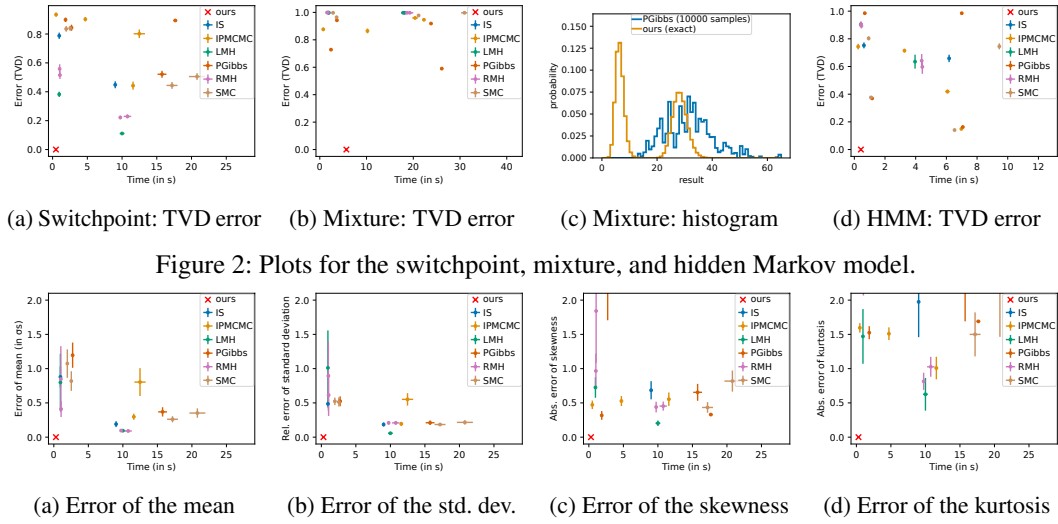

(a) Switchpoint: TVD error    (b) Mixture: TVD error    (c) Mixture: histogram    (d) HMM: TVD error

Figure 2: Plots for the switchpoint, mixture, and hidden Markov model.

(a) Error of the mean    (b) Error of the std. dev.    (c) Error of the skewness    (d) Error of the kurtosis

Figure 3: Comparison of moments for the switchpoint model: approximate inference vs our method.

of these modifications can be handled by [26] or [27]. The results of the first modification (Fig. 1b) are very similar. The more complex second modification takes longer to solve exactly, but less time than approximate inference with 10000 samples, and achieves zero error.

**Switchpoint model**    Our second benchmark is Bayesian switchpoint analysis, which is about detecting a change in the frequency of certain events over time. We use the model from [23] with continuous priors and its 111 real-world data points about the frequency of coal-mining accidents. We compare both the moment errors (Fig. 3) and the TVD errors (Fig. 2a). In both cases, the approximations are less accurate and take longer than our exact method.

**Mixture model**    We consider a binary mixture model on the same data set, with equal mixture weights and a geometric prior for the rates: each data point is observed from a mixture of two Poisson distributions with different rates and the task is to infer these rates. Due to their multimodality, mixture models are notoriously hard for approximate inference methods, which is confirmed in Fig. 2b. Even the runs with the lowest error cover only one of the two modes (cf. the sample histogram in Fig. 2c).

**Hidden Markov model**    We use a hidden Markov model based on [22, Section 2.2], but involving infinite (geometric) priors. It is a two-state system with known transition probabilities and the rate for the observed data depends on the hidden state. We run this model on 30 simulated data points. For this model as well, our method clearly outperforms approximate methods (Fig. 2d).

To our knowledge, our approach is the first to find an exact solution to these problems, except the very first problem without the modifications, which appeared in [26]. For brevity, we only presented the most important aspects of these benchmarks, relegating their encoding as probabilistic programs to Appendix D.3. Code and reproduction instructions are provided in the supplementary material.

## 6 Conclusion

By leveraging generating functions, we have developed and proven correct a framework for exact Bayesian inference on discrete models, even with infinite support and continuous priors. We have demonstrated competitive performance on a range of models specified in an expressive probabilistic programming language, which our tool Genfer processes automatically.

**Future work**    It is a natural question how our method could be integrated with a more general probabilistic programming system. For example, in a sampling-based inference algorithm, one could imagine using generating functions to solve subprograms exactly if this is possible. More generally, it would be desirable to explore how the compositionality of the GF translation can be improved. As it stands, our GF translation describes the joint distribution of all program variables – we never reason "locally" about a subset of the variables. A compositional approach would likely facilitate the application of the GF method to functional probabilistic languages like Anglican.

## Acknowledgments and Disclosure of Funding

We would like to thank Maria Craciun, Hugo Paquet, Tim Reichelt, and Dominik Wagner for providing valuable input on this work. We are especially grateful to Mathieu Huot for insightful discussions and very helpful feedback on a draft of this paper.

This research was supported by the Engineering and Physical Sciences Research Council (studentship 2285273, grant EP/T006579) and the National Research Foundation, Singapore, under its RSS Scheme (NRF-RSS2022-009). For the purpose of Open Access, the authors have applied a CC BY public copyright license to any Author Accepted Manuscript version arising from this submission.

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

# A  Details on the Probabilistic Programming Language SGCL

## A.1  A minimal grammar

A minimal grammar of SGCL looks like this:

$$P ::= \mathsf{skip} \mid P_1; P_2 \mid X_k := a_1 X_1 + \cdots + a_n X_n + c \mid \mathsf{if}\ X_k \in A\ \{P_1\}\ \mathsf{else}\ \{P_2\}$$
$$\mid \mathsf{fail} \mid X_k \sim D \mid X_k \sim D(X_j)$$

where $a_1, \ldots, a_n, c \in \mathbb{R}_{\geq 0}$, $p \in [0,1]$ are literals, and $A$ is a finite subset of $\mathbb{N}$. The $X_k$ for $k = 1, \ldots, n$ are variables that can take on values in $\mathbb{R}_{\geq 0}$. To explain, skip leaves everything unchanged, $P_1; P_2$ runs $P_1$ and then $P_2$, and $X_k := \mathbf{a}^\top \mathbf{X} + c$ performs an affine transformation where $\mathbf{a} \in \mathbb{R}_{\geq 0}^n$ and $c \in \mathbb{R}_{\geq 0}$ (to ensure that the support of $X_k$ is a subset of $\mathbb{R}_{\geq 0}$). The if construct allows branching based on comparisons of a variable $X_k$ with constant natural numbers, with the requirement that $X_k$ be a discrete variable on $\mathbb{N}$, and $X_k \sim D$ samples $X_k$ from a primitive distribution (with constant parameters)

$$\begin{aligned}
\mathcal{D} = \{ &\mathsf{Bernoulli}(p), \mathsf{Categorical}(p_0, \ldots, p_m), \mathsf{Binomial}(m, p), \mathsf{Uniform}\{l..m\}, \\
&\mathsf{NegBinomial}(m, p), \mathsf{Geometric}(p), \mathsf{Poisson}(\lambda), \\
&\mathsf{Exponential}(\alpha), \mathsf{Gamma}(\alpha, \beta), \mathsf{Uniform}[a, b] \\
&\mid p \in [0,1], \alpha, \beta \in \mathbb{R}_{\geq 0}, l, m \in \mathbb{N}, a, b \in \mathbb{R}_{\geq 0}, a \leq b \}
\end{aligned}$$

whereas $X_k \sim D(X_j)$ samples $X_k$ from a compound distribution (with a parameter that is again a variable)

$$\begin{aligned}
\mathcal{D}(X_j) = \{ &\mathsf{Binomial}(X_j, p), \mathsf{NegBinomial}(X_j, p), \mathsf{Poisson}(\lambda \cdot X_j), \mathsf{Bernoulli}(X_j) \\
&\mid p \in [0,1], j \in \{1, \ldots, n\}, \lambda \in \mathbb{R}_{\geq 0} \}.
\end{aligned}$$

## A.2  Syntactic sugar

For convenience, we add some syntactic sugar, which is expanded as follows:

$$\begin{aligned}
X_k \mathrel{+}= \mathbf{a}^\top \mathbf{X} + c &\rightsquigarrow \mathbf{X}_k := (\mathbf{a}[k \mapsto a_k + 1])^\top \mathbf{X} + c \\
X_k \mathrel{+}{\sim} D &\rightsquigarrow X_{n+1} \sim D; X_k \mathrel{+}= X_{n+1} \quad (X_{n+1}\ \text{fresh}) \\
\mathsf{observe}\ \phi &\rightsquigarrow \mathsf{if}\ \phi\ \{\mathsf{skip}\}\ \mathsf{else}\ \{\mathsf{fail}\} \\
\mathsf{if}\ m \sim D\ \{P_1\}\ \mathsf{else}\ \{P_2\} &\rightsquigarrow X_{n+1} \sim D; \mathsf{if}\ X_{n+1} = m\ \{P_1\}\ \mathsf{else}\ \{P_2\} \quad (X_{n+1}\ \text{fresh}) \\
\mathsf{if}\ X_k = m\ \{P_1\}\ \mathsf{else}\ \{P_2\} &\rightsquigarrow \mathsf{if}\ X_k \in \{m\}\ \{P_1\}\ \mathsf{else}\ \{P_2\} \\
\mathsf{if}\ \neg\phi\ \{P_1\}\ \mathsf{else}\ \{P_2\} &\rightsquigarrow \mathsf{if}\ \phi\ \{P_2\}\ \mathsf{else}\ \{P_1\} \\
\mathsf{if}\ \phi_1 \wedge \phi_2\ \{P_1\}\ \mathsf{else}\ \{P_2\} &\rightsquigarrow \mathsf{if}\ \phi_1\ \{\mathsf{if}\ \phi_2\ \{P_1\}\ \mathsf{else}\ \{P_2\}\}\ \mathsf{else}\ \{P_2\} \\
\mathsf{if}\ \phi_1 \vee \phi_2\ \{P_1\}\ \mathsf{else}\ \{P_2\} &\rightsquigarrow \mathsf{if}\ \phi_1\ \{P_1\}\ \mathsf{else}\ \{\mathsf{if}\ \phi_2\ \{P_1\}\ \mathsf{else}\ \{P_2\}\}
\end{aligned}$$

## A.3  Standard measure semantics

**Notation and Conventions**  Throughout the text, we always assume that $\mathbb{N}$ carries the discrete $\sigma$-algebra $\mathcal{P}(\mathbb{N})$ and $\mathbb{R}$ the Borel $\sigma$-algebra $\mathcal{B}(\mathbb{R})$. Products of these spaces (e.g. $\mathbb{N} \times \mathbb{R}$) are endowed with the product $\sigma$-algebra. In the following, we write $\mathbf{x}$ for $(x_1, \ldots, x_n)$ and use the update notation $\mathbf{x}[i \mapsto a]$ to denote $(x_1, \ldots, x_{i-1}, a, x_{i+1}, \ldots, x_n)$. In fact, we even write updates $\mathbf{x}[i_1 \mapsto a_1, \ldots, i_k \mapsto a_k]$, which stands for $\mathbf{x}[i_1 \mapsto a_1] \cdots [i_k \mapsto a_k]$. Finally, we write $\mathbf{1}_k$ for the $k$-tuple $(1, \ldots, 1)$ and similarly for $\mathbf{0}_k$. We write $\mathbf{0}$ for the zero measure or the constant zero function. We also use Iverson brackets $[\phi]$ for a logical condition $\phi$: $[\phi]$ is 1 if $\phi$ is satisfied and 0 otherwise. For example, the indicator function $\mathbf{1}_S$ of a set $S$ can be defined using Iverson brackets: $\mathbf{1}_S(x) = [x \in S]$. For a measure $\mu$ on $\mathbb{R}^n$, we define $\mu_{X_k \in A}(S) = \mu(\{\mathbf{X} \in S \mid X_k \in A\})$.

**Standard transformer semantics**  The standard semantics for such a probabilistic program uses distribution transformers [19]. Since our language includes conditioning, the distribution of program states will not always be a probability distribution, but a subprobability distribution $\mu$ on $\mathbb{R}_{\geq 0}^n$ for $n$ variables, i.e. a function $\mu : \mathcal{B}(\mathbb{R}_{\geq 0}^n) \to [0,1]$ with $\mu(\mathbb{R}_{\geq 0}^n) \leq 1$. The semantics of a program $P$ is thus a measure transformer $[\![P]\!]_{\mathsf{std}} : \mathrm{Meas}(\mathbb{R}_{\geq 0}^k) \to \mathrm{Meas}(\mathbb{R}_{\geq 0}^k)$. We assume that the initial measure

$$\llbracket \mathsf{skip} \rrbracket_{\mathsf{std}}(\mu) = \mu$$

$$\llbracket P_1; P_2 \rrbracket_{\mathsf{std}}(\mu) = \llbracket P_2 \rrbracket_{\mathsf{std}}(\llbracket P_1 \rrbracket_{\mathsf{std}}(\mu))$$

$$\llbracket X_k := \mathbf{a}^\top \boldsymbol{X} + c \rrbracket_{\mathsf{std}}(\mu)(S) = \mu(\{\boldsymbol{X} \in \mathbb{R}^n \mid \boldsymbol{X}[k \mapsto \mathbf{a}^\top \boldsymbol{X} + c] \in S\})$$

$$\llbracket \mathsf{if}\ X_k \in A\ \{P_1\}\ \mathsf{else}\ \{P_2\} \rrbracket_{\mathsf{std}}(\mu) = \llbracket P_1 \rrbracket_{\mathsf{std}}(\mu_{X_k \in A}) + \llbracket P_2 \rrbracket_{\mathsf{std}}(\mu - \mu_{X_k \in A})$$

$$\text{where } \mu_{X_k \in A}(S) = \mu(\{\boldsymbol{X} \in S \mid X_k \in A\})$$

$$\llbracket X_k \sim D \rrbracket_{\mathsf{std}}(\mu)(S) = \int \int [\boldsymbol{X}[k \mapsto Y] \in S]\, \mathrm{d}D(Y)\, \mathrm{d}\mu(\boldsymbol{X})$$

$$\llbracket X_k \sim D(X_j) \rrbracket_{\mathsf{std}}(\mu)(S) = \int \int [\boldsymbol{X}[k \mapsto Y] \in S]\, \mathrm{d}D(X_j)(Y)\, \mathrm{d}\mu(\boldsymbol{X})$$

$$\llbracket \mathsf{fail} \rrbracket_{\mathsf{std}}(\mu) = \mathbf{0}$$

$$\mathsf{normalize}(\mu) = \frac{\mu}{\int \mathrm{d}\mu(\boldsymbol{X})}$$

Figure 4: The standard measure semantics $\llbracket - \rrbracket_{\mathsf{std}}$.

is the probability distribution where every variable $X_k$ is 0 with probability 1, i.e. $\boldsymbol{X} \sim \mathsf{Dirac}(\mathbf{0}_n)$. The semantics is shown in Fig. 4 where $\boldsymbol{X}$ is a vector of length $n$.

The first 5 rules were essentially given by [19], with slightly different notation and presentation. Kozen defines some of the measures on rectangles ($S = S_1 \times \cdots \times S_n$) only, which uniquely extends to a measure on all sets in the product $\sigma$-algebra. By contrast, we directly define the measure on all sets in the $\sigma$-algebra using integrals.

We quickly explain the rules. The skip command does nothing, so the distribution $\mu$ of the program is unchanged. Chaining commands $P_1; P_2$ has the effect of composing the corresponding trans-formations of $\mu$. Affine transformations of program variables yield a measure of $S$ that is given by the original measure of the preimage of the event $S$ under the affine map. For conditionals if $X_k \in A\ \{P_1\}\ \mathsf{else}\ \{P_2\}$, we consider only outcomes satisfying $X_k \in A$ in the first branch and the complement in the second branch and then sum both possibilities. Sampling from distributions $X_k \sim D$ is a bit more complicated. Note that $\mu(S) = \int [\boldsymbol{X} \in S]\, \mathrm{d}\mu(\boldsymbol{X})$. To obtain the measure after sampling $X_k \sim D$, we care about the new state $\boldsymbol{X}[k \mapsto Y]$ being in $S$, where $Y$ has distribution $D$. This is exactly expressed by the double integral. The rule for compound distributions $X_k \sim D(X_j)$ is essentially the same as the one for distributions with constant parameters $X_k \sim D$, except the dependency on $X_j$ needs to be respected.

The two new rules (not derivable from Kozen's work) are for fail and the implicit normalization normalize$(\mu)$ as a final step at the end of a program. The rule for fail departs from Kozen's semantics in the sense that it yields a subprobability distribution (i.e. the total mass can be less than 1). Intuitively, fail "cuts off" or assigns the probability 0 to certain branches in the program. This is called *hard conditioning* in the PPL literature. For example, if $X_1 = 5\ \{\mathsf{skip}\}\ \mathsf{else}\ \{\mathsf{fail}\}$ has the effect of conditioning on the event that $X_1 = 5$ because all other scenarios are assigned probability 0.

The normalization construct corresponds to the division by the evidence in Bayes' rule and scales the subprobability distribution of a subprogram $P$ back to a probability distribution. The semantics is $\frac{\mu}{\int \mathrm{d}\mu(\boldsymbol{X})}$, so the distribution $\mu$ is scaled by the inverse of the total probability mass of $\mu$, thus normalizing it.

## B   Details on Generating Functions

An exhaustive list of the supported distributions and their generating functions can be found in Tables 4a and 4b.

(a) GFs for common distributions

| Distribution $D$ | $\mathsf{gf}(X \sim D)(x)$ |
|---|---|
| Dirac($a$) | $x^a$ |
| Bernoulli($p$) | $px + 1 - p$ |
| Categorical($p_0, \ldots, p_n$) | $\sum_{i=0}^{n} p_i x^i$ |
| Binomial($n, p$) | $(px + 1 - p)^n$ |
| Uniform$\{a..b\}$ | $\frac{x^a - x^{b+1}}{(b-a+1)(1-x)}$ |
| NegBinomial($r, p$) | $\left(\frac{p}{1-(1-p)x}\right)^r$ |
| Geometric($p$) | $\frac{p}{1-(1-p)x}$ |
| Poisson($\lambda$) | $e^{\lambda(x-1)}$ |
| Exponential($\lambda$) | $\frac{\lambda}{\lambda - \log x}$ |
| Gamma($\alpha, \beta$) | $\left(\frac{\beta}{\beta - \log x}\right)^\alpha$ |
| Uniform$[a, b]$ | $\begin{cases} \frac{x^b - x^a}{(b-a)\log x} & x \neq 1 \\ 1 & x = 1 \end{cases}$ |

(b) GFs for common compound distributions

| Distribution $D(Y)$ | $\mathsf{gf}(X \sim D(Y))(x)$ |
|---|---|
| Binomial($Y, p$) | $\mathsf{gf}(Y)(1 - p + px)$ |
| NegBinomial($Y, p$) | $\mathsf{gf}(Y)\left(\frac{p}{1-(1-p)x}\right)$ |
| Poisson($\lambda \cdot Y$) | $\mathsf{gf}(Y)(e^{\lambda(x-1)})$ |
| Bernoulli($Y$) | $1 + (x - 1) \cdot (\mathsf{gf}(Y))'(1)$ |

$$[\![\mathsf{skip}]\!]_{\mathsf{gf}}(G) = G$$

$$[\![P_1; P_2]\!]_{\mathsf{gf}}(G) = [\![P_2]\!]_{\mathsf{gf}}([\![P_1]\!]_{\mathsf{gf}}(G))$$

$$[\![X_k := \mathbf{a}^\top \boldsymbol{X} + c]\!]_{\mathsf{gf}}(G)(\mathbf{x}) = x_k^c \cdot G(\boldsymbol{x}')$$

$$\text{where } x_k' := x_k^{a_k} \text{ and } x_i' := x_i x_k^{a_i} \text{ for } i \neq k$$

$$[\![\mathsf{if}\ X_k \in A\ \{P_1\}\ \mathsf{else}\ \{P_2\}]\!]_{\mathsf{gf}}(G) = [\![P_1]\!]_{\mathsf{gf}}(G_{X_k \in A}) + [\![P_2]\!]_{\mathsf{gf}}(G - G_{X_k \in A})$$

$$\text{where } G_{X_k \in A}(\mathbf{x}) = \sum_{i \in A} \frac{\partial_k^i G(\mathbf{x}[k \mapsto 0])}{i!} x_k^i \quad \text{and } A \subset \mathbb{N} \text{ finite}$$

$$\text{if } X_k \text{ has support in } \mathbb{N}$$

$$[\![X_k \sim D]\!]_{\mathsf{gf}}(G)(\mathbf{x}) = G(\mathbf{x}[k \mapsto 1])\mathsf{gf}(D)(x_k)$$

$$[\![X_k \sim D(X_j)]\!]_{\mathsf{gf}}(G)(\mathbf{x}) = G(\mathbf{x}[k \mapsto 1, j \mapsto x_j \cdot \mathsf{gf}(D(1))(x_k)])$$

$$\text{for } D \in \{\mathsf{Binomial}(-, p), \mathsf{NegBinomial}(-, p), \mathsf{Poisson}(\lambda \cdot -)\}$$

$$[\![X_k \sim \mathsf{Bernoulli}(X_j)]\!]_{\mathsf{gf}}(G)(\mathbf{x}) = G(\boldsymbol{x}[k \mapsto 1]) + x_j(x_k - 1) \cdot \partial_j G(\boldsymbol{x}[k \mapsto 1])$$

$$\text{if } X_j \text{ has support in } [0, 1]$$

$$[\![\mathsf{fail}]\!]_{\mathsf{gf}}(G) = \mathbf{0}$$

$$\mathsf{normalize}(G) = \frac{G}{G(\mathbf{1}_n)}$$

Figure 5: The generating function (GF) semantics $[\![-]\!]_{\mathsf{gf}}$.

## B.1 Generating function semantics

The full generating function semantics can be found in Fig. 5. The first five rules in Fig. 5 were presented in [17, 4]. The GF for sampling from $\mathsf{Binomial}(X_j, p)$ was first given in [26]. Chen et al. [4] also implicitly describe the GF semantics of sampling from $\mathsf{Binomial}(X_j, p)$ and $\mathsf{NegBinomial}(X_j, p)$ by expressing it as summing $X_j$ iid samples from $\mathsf{Bernoulli}(p)$ and $\mathsf{Geometric}(p)$, respectively.

Let us discuss each rule in detail. The skip command leaves the distribution and hence the generating function unchanged. Chaining commands $P_1; P_2$ is again achieved by composing the individual transformers for each command. To obtain some intuition for the affine transformation case $X_k := \mathbf{a}^\top \mathbf{X} + c$, suppose we only have 2 variables: $X_1 := 2X_1 + 3X_2 + 5$. The idea is that the transformed generating function is given by $[\![X_1 := 2X_1 + 3X_2 + 5]\!]_{\mathsf{gf}}(G)(\mathbf{x}) = \mathbb{E}[x_1^{2X_1 + 3X_2 + 5} x_2^{X_2}] = x_1^5 \mathbb{E}[(x_1^2)^{X_1}(x_1^3 x_2)^{X_2}] = x_1^5 \cdot G(x_1^2, x_1^3 x_2)$.

The semantics of if $X_k \in A \{P_1\}$ else $\{P_2\}$ uses $G_{X_k \in A}$. If $G$ is the generating function for a measure $\mu$ then $G_{X_k \in A}$ is the generating function for the measure $\mu_{X_k \in A}(S) := \mu(\{\mathbf{x} \in S \mid x_k \in A\})$. To get intuition for this, remember that for discrete variables $X_1, X_2$, we have $G(x_1, x_2) = \sum_{X_1, X_2} \mu(\{(X_1, X_2)\}) \cdot x_1^{X_1} x_2^{X_2}$. If we want to obtain the generating function for $X_1 = 5$, we want to keep only terms where $x_1$ has exponent 5. In other words, we need the Taylor coefficient $\frac{\partial_1^5 G(0, x_2)}{5!}$. This should give some intuition for $G_{X_k \in A}$. The semantics of if $X_k \in A \{P_1\}$ else $\{P_2\}$ transforms $G_{X_k \in A}$ in the then-branch and the complement $G - G_{X_k \in A}$ in the else branch, summing the two possibilities.

For sampling, say $X_1 \sim D$, we first marginalize out the old $X_1$ by substituting 1 for $x_1$ in $G$: $\mathbb{E}[x_2^{X_2}] = \mathbb{E}[1^{X_1} x_2^{X_2}] = G(1, x_2)$. Then we multiply with the GF of the new distribution $D$. The rules for compound distributions $D(X_j)$ are more involved, so we offer the following intuition.

For the compound distributions $D(n) = \mathsf{Binomial}(n, p), \mathsf{NegBinomial}(n, p), \mathsf{Poisson}(\lambda \cdot n)$, we make essential use of their property that $\mathsf{gf}(D(n))(x) = (\mathsf{gf}(D(1))(x))^n$. So if $G(x) := \mathbb{E}[x^X]$ is the GF of $X$ and $Y \sim D(X)$, then $\mathbb{E}_{X,Y}[x^X y^Y] = \mathbb{E}_X[x^X \mathbb{E}_{Y \sim D(X)}[y^Y]] = \mathbb{E}_X[x^X \mathsf{gf}(D(X))(y)] = \mathbb{E}_X[x^X(\mathsf{gf}(D(1))(y))^X] = \mathbb{E}[(x \cdot \mathsf{gf}(D(1))(y))^X] = G(x \cdot \mathsf{gf}(D(1))(y))$. This is not a fully formal argument, but it should give some intuition for the rule.

For $D(p) = \mathsf{Bernoulli}(p)$, the intuition is that if $G(x) := \mathbb{E}[x^X]$ is the GF of $X$ and if $Y \sim \mathsf{Bernoulli}(X)$, then $H(x, y) = \mathbb{E}[x^X y^Y] = \mathbb{E}[x^X((1 - X)y^0 + Xy^1)] = \mathbb{E}[x^X - (y - 1)Xx^X] = G(x) - (y - 1)xG'(x)$.

The GF semantics of fail is the zero function because its distribution is zero everywhere. For normalization, note that substituting $\mathbf{1}_n$ in the generating function computes the marginal probabilities, similarly to the substitution of 1 in the semantics of sampling. It scales the GF of the subprogram $P$ by the inverse of the normalizing constant obtained by that substitution.

We hope these explanations give some intuition for the GF semantics, but of course, proof is required.

## B.2 Correctness proof

For the correctness proof, we need to describe the set of $\mathbf{x} \in \mathbb{R}^n$ where the generating function is defined. For $R > 0$ and a measure $\mu$ on $\mathbb{R}^n_{\geq 0}$, define it to be the product of open intervals

$$T(R, \mu) := (Q_1, R) \times \cdots \times (Q_n, R)$$

where $Q_i = -R$ if the $i$-th component of $\mu$ is supported on $\mathbb{N}$ (i.e. $\mu(S) = \mu(\{\mathbf{X} \in S \mid X_i \in \mathbb{N}\})$ for any measurable $S$) and $Q_i = 0$ otherwise. The case split is necessary because the generating functions are better-behaved, and thus have a larger domain, if the measure is supported on $\mathbb{N}$. In fact, it is important that $x_i = 0$ is possible in the $\mathbb{N}$-case because otherwise we would not be able to extract probability masses from the GF.

**Theorem B.1** (Correctness). *The GF semantics is correct w.r.t. the standard semantics $[\![-]\!]_{\mathsf{std}}$: for any SGCL program $P$ and subprobability distribution $\mu$ on $\mathbb{R}^n_{\geq 0}$, we have $[\![P]\!]_{\mathsf{gf}}(\mathsf{gf}(\mu)) = \mathsf{gf}([\![P]\!]_{\mathsf{std}}(\mu))$. In particular, it correctly computes the GF of the posterior distribution of $P$ as $\mathsf{normalize}([\![P]\!]_{\mathsf{gf}}(\mathbf{1}))$.*

*If $\mathsf{gf}(\mu)$ is defined on $T(R, \mu)$ for some $R > 1$ then $[\![P]\!]_{\mathsf{gf}}(\mathsf{gf}(\mu))$ is defined on $T(R', [\![P]\!]_{\mathsf{std}}(\mu))$ for some $R' > 1$. In particular, the GFs $[\![P]\!]_{\mathsf{gf}}(\mathbf{1})$ and thus $\mathsf{normalize}([\![P]\!]_{\mathsf{gf}}(\mathbf{1}))$ are defined on $T(R, [\![P]\!]_{\mathsf{std}}(\mu))$ for some $R > 1$.*

*Proof.* As usual for such statements, we prove this by induction on the structure of the program. The program $P$ can take one of the following forms:

**Skip.** For the trivial program, we have $[\![P]\!]_{\text{gf}}(G) = G$ and $[\![P]\!]_{\text{std}}(\mu) = \mu$, so the claim is trivial.

**Chaining.** If $P$ is $P_1; P_2$, we know by the inductive hypothesis that $[\![P_1]\!]_{\text{gf}}(\text{gf}(\mu)) = \text{gf}([\![P_1]\!]_{\text{std}}(\mu))$ and similarly for $P_2$. Taking this together, we find by the definitions of the semantics:

$$[\![P_1; P_2]\!]_{\text{gf}}(\text{gf}(\mu)) = [\![P_2]\!]_{\text{gf}}([\![P_1]\!]_{\text{gf}}(\text{gf}(\mu))) = [\![P_2]\!]_{\text{gf}}(\text{gf}([\![P_1]\!]_{\text{std}}(\mu)))$$
$$= \text{gf}([\![P_2]\!]_{\text{std}}([\![P_1]\!]_{\text{std}}(\mu))) = \text{gf}([\![P_1; P_2]\!]_{\text{std}}(\mu)).$$

The claim about the domain of $[\![P]\!]_{\text{gf}}(G)$ follows directly from the inductive hypotheses.

**Affine assignments.** If $P$ is $X_k := \mathbf{a}^\top \mathbf{X} + c$, we have $[\![P]\!]_{\text{std}}(\mu)(S) = \mu(\{\mathbf{X} \in \mathbb{R}^n \mid \mathbf{X}[k \mapsto \mathbf{a}^\top \mathbf{X} + c] \in S\})$ and thus

$$\text{gf}([\![P]\!]_{\text{std}}(\mu))(\boldsymbol{x}) = \int \boldsymbol{x}^{\mathbf{X}} \, \text{d}([\![P]\!]_{\text{std}}(\mu))(\mathbf{X})$$

$$= \int x_1^{X_1} \cdots x_{k-1}^{X_{k-1}} x_k^{a_1 X_1 + \cdots + a_n X_n + c} x_{k+1}^{X_{k+1}} \cdots x_n^{X_n} \, \text{d}\mu(X_1, \ldots, X_n)$$

$$= x_k^c \cdot \int (x_1 x_k^{a_1})^{X_1} \cdots (x_{k-1} x_k^{a_{k-1}})^{X_{k-1}} (x_k^{a_k})^{X_k} \cdot$$
$$(x_{k+1} x_k^{a_{k+1}})^{X_{k+1}} \cdots (x_n x_k^{a_n})^{X_n} \, \text{d}\mu(\mathbf{X})$$

$$= x_k^c \cdot \text{gf}(\mu)(x_1 x_k^{a_1}, \ldots, x_{k-1} x_k^{a_{k-1}}, x_k^{a_k}, x_{k+1} x_k^{a_{k+1}}, \ldots, x_n x_k^{a_n})$$

$$= [\![P]\!]_{\text{gf}}(\text{gf}(\mu))(\boldsymbol{x})$$

The claim that it is defined on some $T(R', [\![P]\!]_{\text{std}}(\mu))$ holds if we choose $R' = \min(\sqrt{R}, \min_{i=1}^n \sqrt[2a_i]{R}) > 1$ because then $|x_i x_k^{a_i}| < \sqrt{R}(\sqrt[2a_i]{R})^{a_i} \leq R$ for $i \neq k$ and $|x_k^{a_k}| < (\sqrt[2a_k]{R})^{a_k} \leq \sqrt{R} \leq R$.

**Conditionals.** If $P$ is if $X_k \in A \{P_1\}$ else $\{P_2\}$, the GF semantics defines $(\text{gf}(\mu))_{X_k \in A}$, which is the same as $\text{gf}(\mu_{X_k \in A})$ by Lemma B.2. Using this fact, the linearity of $\text{gf}(-)$, and the induction hypothesis, we find

$$\text{gf}([\![P]\!]_{\text{std}}(\mu)) = \text{gf}([\![P_1]\!]_{\text{std}}(\mu_{X_k \in A})) + \text{gf}([\![P_2]\!]_{\text{std}}(\mu - \mu_{X_k \in A}))$$
$$= [\![P_1]\!]_{\text{gf}}(\text{gf}(\mu_{X_k \in A})) + [\![P_2]\!]_{\text{gf}}(\text{gf}(\mu) - \text{gf}(\mu_{X_k \in A}))$$
$$= [\![P_1]\!]_{\text{gf}}((\text{gf}(\mu))_{X_k \in A}) + [\![P_2]\!]_{\text{gf}}(\text{gf}(\mu) - (\text{gf}(\mu))_{X_k \in A})$$
$$= [\![P]\!]_{\text{gf}}(\text{gf}(\mu))$$

Also by Lemma B.2, we know that $(\text{gf}(\mu))_{X_k \in A}$ (and thus $(\text{gf}(\mu) - (\text{gf}(\mu))_{X_k \in A})$) is defined on $T(R, \mu)$, so by the induction hypothesis, both $[\![P_1]\!]_{\text{gf}}((\text{gf}(\mu))_{X_k \in A})$ and $[\![P_2]\!]_{\text{gf}}(\text{gf}(\mu) - \text{gf}(\mu_{X_k \in A}))$ are defined on some $T(R')$. Hence the same holds for $[\![P]\!]_{\text{gf}}(\text{gf}(\mu))$.

**Sampling.** If $P$ is $X_k \sim D$, we find:

$$\text{gf}([\![P]\!]_{\text{std}}(\mu))(\boldsymbol{x}) = \int \boldsymbol{x}^{\mathbf{X}} \, \text{d}([\![P]\!]_{\text{std}}(\mu))(\mathbf{X})$$

$$= \int \int \boldsymbol{x}^{\mathbf{X}[k \mapsto Y]} \, \text{d}D(Y) \, \text{d}\mu(\mathbf{X})$$

$$= \int (\boldsymbol{x}[k \mapsto 1])^{\mathbf{X}} \int x_k^Y \, \text{d}D(Y) \, \text{d}\mu(\mathbf{X})$$

$$= \int (\boldsymbol{x}[k \mapsto 1])^{\mathbf{X}} \, \text{d}\mu(\mathbf{X}) \cdot \int x_k^Y \, \text{d}D(Y)$$

$$= \text{gf}(\mu)(\boldsymbol{x}[k \mapsto 1])\text{gf}(D)(x_k)$$

$$= [\![P]\!]_{\text{gf}}(\text{gf}(\mu))(\boldsymbol{x})$$

Regarding the domain of $[\![P]\!]_{\text{gf}}(\text{gf}(\mu))$, note that $\text{gf}(D)$ is defined on all of $\mathbb{R}$ for any finite distribution and for the Poisson distribution (cf. Table 4a). For $D \in \{\text{Geometric}(p), \text{NegBinomial}(n, p)\}$, the GF $\text{gf}(D)$ is defined on $(-\frac{1}{1-p}, \frac{1}{1-p})$, so we can pick $R' := \min(R, \frac{1}{1-p}) > 1$. The GF of

Exponential($\lambda$) is defined for $\log x < \lambda$, i.e. $x < \exp(\lambda)$, so we can pick $R' := \min(R, e^\lambda) > 1$. The GF of Gamma($\alpha, \beta$) is defined for $\log x < \beta$, i.e. $x < \exp(\beta)$, so we can pick $R' := \min(R, e^\beta) > 1$. The GF of Uniform$[a, b]$ is defined on $\mathbb{R}$, so $R' := R$ works.

**Compound distributions.** If $P$ is $X_k \sim D(X_j)$, we use the fact that $\mathsf{gf}(D(n))(\boldsymbol{x}) = (\mathsf{gf}(D(1))(\boldsymbol{x}))^n$ for $D(n) \in \{\text{Binomial}(n, p), \text{NegBinomial}(n, p), \text{Poisson}(\lambda \cdot n)\}$, which can easily be checked by looking at their generating functions (cf. Table 4b).

$$
\begin{aligned}
\mathsf{gf}(\llbracket P \rrbracket_{\mathsf{std}}(\mu))(\boldsymbol{x}) &= \int \boldsymbol{x}^{\boldsymbol{X}} \, \mathrm{d}(\llbracket P \rrbracket_{\mathsf{std}}(\mu))(\boldsymbol{X}) \\
&= \int \int \boldsymbol{x}^{\boldsymbol{X}[k \mapsto Y]} \, \mathrm{d}D(X_j)(Y) \, \mathrm{d}\mu(\boldsymbol{X}) \\
&= \int (\boldsymbol{x}[k \mapsto 1])^{\boldsymbol{X}} \int x_k^Y \, \mathrm{d}D(X_j)(Y) \, \mathrm{d}\mu(\boldsymbol{X}) \\
&= \int (\boldsymbol{x}[k \mapsto 1])^{\boldsymbol{X}} \mathsf{gf}(D(X_j))(x_k) \, \mathrm{d}\mu(\boldsymbol{X}) \\
&= \int (\boldsymbol{x}[k \mapsto 1])^{\boldsymbol{X}} (\mathsf{gf}(D(1))(x_k))^{X_j} \, \mathrm{d}\mu(\boldsymbol{X}) \\
&= \int (\boldsymbol{x}[k \mapsto 1, j \mapsto x_j \cdot \mathsf{gf}(D(1))(x_k)])^{\boldsymbol{X}} \, \mathrm{d}\mu(\boldsymbol{X}) \\
&= \mathsf{gf}(\mu)(\boldsymbol{x}[k \mapsto 1, j \mapsto x_j \cdot \mathsf{gf}(D(1))(x_k)]) \\
&= \llbracket P \rrbracket_{\mathsf{gf}}(\mathsf{gf}(\mu))(\boldsymbol{x})
\end{aligned}
$$

Regarding the domain of $\llbracket P \rrbracket_{\mathsf{gf}}(\mathsf{gf}(\mu))$, we have to ensure that $|x_j \cdot \mathsf{gf}(D(1))(x_k)| < R$. For the Binomial$(1, p)$ distribution, we choose $R' = \sqrt{R} > 1$ such that $|x_j||1 - p + px_k| < \sqrt{R}(1 - p + p\sqrt{R}) \leq \sqrt{R}((1-p)\sqrt{R} + p\sqrt{R}) < R$. For the NegBinomial$(1)$ distribution, we choose $R' = \min(\sqrt{R}, \frac{\sqrt{R}-p}{\sqrt{R}(1-p)}) = \min(\sqrt{R}, 1 + \frac{p(\sqrt{R}-1)}{\sqrt{R}-p\sqrt{R}}) > 1$ because for any $x \in (-R', R)$, we have

$$
\left| \frac{p}{1 - (1-p)x_j} \right| < \frac{p}{1 - (1-p)R'} \leq \frac{p}{1 - \frac{\sqrt{R}-p}{\sqrt{R}}} = \frac{p\sqrt{R}}{p} = \sqrt{R}
$$

and thus $\left| x_j \frac{p}{1-(1-p)x_k} \right| < R$. For the Poisson$(\lambda)$ distribution, we choose $R' = \min(\sqrt{R}, 1 + \frac{\log(R)}{2\lambda}) > 1$ because then $|x_j \cdot \exp(\lambda(x_k - 1))| < \sqrt{R}\exp(\lambda\frac{\log(R)}{2\lambda}) = R$.

For the Bernoulli distribution $D(X_j) = \text{Bernoulli}(X_j)$, we reason as follows:

$$
\begin{aligned}
\mathsf{gf}(\llbracket P \rrbracket_{\mathsf{std}}(\mu))(\boldsymbol{x}) = \cdots &= \int (\boldsymbol{x}[k \mapsto 1])^{\boldsymbol{X}} \mathsf{gf}(D(X_j))(x_k) \, \mathrm{d}\mu(\boldsymbol{X}) \\
&= \int (\boldsymbol{x}[k \mapsto 1])^{\boldsymbol{X}} (1 + X_j(x_k - 1)) \, \mathrm{d}\mu(\boldsymbol{X}) \\
&= \int (\boldsymbol{x}[k \mapsto 1])^{\boldsymbol{X}} \, \mathrm{d}\mu(\boldsymbol{X}) + (x_k - 1) \cdot \int X_j(\boldsymbol{x}[k \mapsto 1])^{\boldsymbol{X}} \, \mathrm{d}\mu(\boldsymbol{X}) \\
&= \mathsf{gf}(\mu)(\boldsymbol{x}[k \mapsto 1]) + (x_k - 1) \cdot \int x_j \cdot \partial_j (\boldsymbol{x}[k \mapsto 1])^{\boldsymbol{X}} \, \mathrm{d}\mu(\boldsymbol{X}) \\
&= \mathsf{gf}(\mu)(\boldsymbol{x}[k \mapsto 1]) + x_j(x_k - 1) \cdot \partial_j \int (\boldsymbol{x}[k \mapsto 1])^{\boldsymbol{X}} \, \mathrm{d}\mu(\boldsymbol{X}) \\
&= \mathsf{gf}(\mu)(\boldsymbol{x}[k \mapsto 1]) + x_j(x_k - 1) \cdot \partial_j \mathsf{gf}(\mu)(\boldsymbol{x}[k \mapsto 1]) \\
&= \llbracket P \rrbracket_{\mathsf{gf}}(\mathsf{gf}(\mu))(\boldsymbol{x})
\end{aligned}
$$

Note that the interchange of integral and differentiation is allowed by Lemma B.3. It also implies that $\llbracket P \rrbracket_{\mathsf{gf}}(\mathsf{gf}(\mu))(\boldsymbol{x})$ is defined for $x_j \in (0, R)$ by Lemma B.3.

**Fail.** If $P$ is fail, then $\llbracket P \rrbracket_{\mathsf{std}}(\mu)$ is the zero measure and $\llbracket P \rrbracket_{\mathsf{gf}}(\mathsf{gf}(\mu))$ is the zero function, so the claim holds trivially. This GF is clearly defined everywhere.

**Normalization.** For normalization, we make use of the linearity of $\mathsf{gf}(-)$.

$$\mathsf{gf}(\mathsf{normalize}(\mu)) = \mathsf{gf}\left(\frac{\mu}{\int \mathrm{d}\mu(\boldsymbol{X})}\right)$$

$$= \frac{\mathsf{gf}(\mu)}{\int \mathrm{d}\mu(\boldsymbol{X})}$$

$$= \frac{\mathsf{gf}(\mu)}{\int 1^{X_1} \cdots 1^{X_n} \mathrm{d}\mu(\boldsymbol{X})}$$

$$= \frac{\mathsf{gf}(\mu)}{\mathsf{gf}(\mu)(\boldsymbol{1}_n)}$$

$$= \mathsf{normalize}(\mathsf{gf}(\mu))$$

Furthermore, if $\mathsf{gf}(\mu)$ is defined on $T(R, \mu)$, then so is $\mathsf{normalize}(\mathsf{gf}(\mu))$ on $T(R, \mathsf{normalize}(\mu)) = T(R, \mu)$.

This finishes the induction and proves the claim. $\qquad\qquad\square$

**Lemma B.2.** *Let $A \subset \mathbb{N}$ be a finite set. Let $\mu$ be a measure on $\mathbb{R}^n$ such that its $k$-th component is supported on $\mathbb{N}$, i.e. $\mu(S) = \mu(\{\boldsymbol{X} \in S \mid X_k \in \mathbb{N}\})$ for all measurable $S \subseteq \mathbb{R}_{\geq 0}$. Define $\mu_{X_k \in A}(S) := \mu(\{\boldsymbol{X} \in S \mid X_k \in A\})$. Let $G := \mathsf{gf}(\mu)$. Then*

$$\mathsf{gf}(\mu_{X_k \in A})(\boldsymbol{x}) = G_{X_k \in A}(\boldsymbol{x}) := \sum_{i \in A} \frac{\partial_k^i G(\boldsymbol{x}[k \mapsto 0])}{i!} \cdot x_k^i$$

*Furthermore, if $G$ is defined on $T(R, \mu)$ then so is $G_{X_k \in A}$ on $T(R, \mu_{X_k \in A}) = T(R, \mu)$.*

*Proof.* Since $\mu_{X_k \in A} = \sum_{i \in A} \mu_{X_k = i}$ where we write $\mu_{X_k = i} := \mu_{X_k \in \{i\}}$, we have:

$$\mathsf{gf}(\mu_{X_k = i})(\boldsymbol{x}) = \int \boldsymbol{x}^{\boldsymbol{X}} \, \mathrm{d}\mu_{X_k = i}(\boldsymbol{X})$$

$$= \int x_1^{X_1} \cdots x_{k-1}^{X_{k-1}} \cdot x_k^{X_k} \cdot x_{k+1}^{X_{k+1}} \cdots x_n^{X_n} \cdot [X_k = i] \, \mathrm{d}\mu(\boldsymbol{X})$$

$$= x_k^i \int x_1^{X_1} \cdots x_{k-1}^{X_{k-1}} \cdot x_{k+1}^{X_{k+1}} \cdots x_n^{X_n} \cdot \left(\frac{1}{i!} \left.\frac{\partial^i x_k^{X_k}}{\partial x_k^i}\right|_{x_k=0}\right) \mathrm{d}\mu(\boldsymbol{X})$$

The reason for the last step is that for the function $f(x) = x^l$, we have

$$f^{(i)}(x) = \begin{cases} 0 & \text{if } l < i \\ x^{l-i} \cdot \prod_{j=l-i+1}^{l} j & \text{if } l \geq i \end{cases}$$

Since $0^{l-i}$ is 0 for $l > i$ and 1 for $l = i$, we find that $f^{(i)}(0) = [i = l] \cdot i!$. Above, we used this fact with $l = X_k$ and $x = x_k$. We continue:

$$\cdots = x_k^i \int \frac{1}{i!} \left.\left(x_1^{X_1} \cdots x_{k-1}^{X_{k-1}} \cdot \frac{\partial^i x_k^{X_k}}{\partial x_k^i} \cdot x_{k+1}^{X_{k+1}} \cdots x_n^{X_n}\right)\right|_{x_k=0} \mathrm{d}\mu(\boldsymbol{X})$$

$$= \int \left.\frac{\partial^i}{\partial x_k^i} \boldsymbol{x}^{\boldsymbol{X}} \, \mathrm{d}\mu(\boldsymbol{X})\right|_{x_k=0} \cdot \frac{x_k^i}{i!}$$

$$= \left.\frac{\partial^i}{\partial x_k^i} \int \boldsymbol{x}^{\boldsymbol{X}} \, \mathrm{d}\mu(\boldsymbol{X})\right|_{x_k=0} \cdot \frac{x_k^i}{i!}$$

$$= \partial_k^i G(\boldsymbol{x}[k \mapsto 0]) \cdot \frac{x_k^i}{i!}$$

as desired. Note that the integral and differentiation operators can be interchanged by Lemma B.3. $\quad\square$

**Lemma B.3.** *If the integral $\int \boldsymbol{x}^{\boldsymbol{X}} \, \mathrm{d}\mu(\boldsymbol{X})$ for a measure $\mu$ on $\mathbb{R}^n_{\geq 0}$ is defined for all $\boldsymbol{x} \in T(R, \mu)$, we have*

$$\partial_{i_1} \cdots \partial_{i_m} \int \boldsymbol{x}^{\boldsymbol{X}} \, \mathrm{d}\mu(\boldsymbol{X}) = \int \partial_{i_1} \cdots \partial_{i_m} \boldsymbol{x}^{\boldsymbol{X}} \, \mathrm{d}\mu(\boldsymbol{X})$$

*and both sides are defined for all $\boldsymbol{x} \in T(R, \mu)$.*

*Furthermore, the right-hand side has the form $\int p(\boldsymbol{X}) \boldsymbol{x}^{\boldsymbol{X}-\boldsymbol{w}} \, \mathrm{d}\mu(\boldsymbol{X})$ for a polynomial $p$ in $n$ variables and $\boldsymbol{w} \in \mathbb{N}^n$, with the property that $p(\boldsymbol{X}) = 0$ whenever $X_j < w_j$ for some $j$, in which case we define $p(\boldsymbol{X}) \boldsymbol{x}^{\boldsymbol{X}-\boldsymbol{w}} := 0$, even if $\boldsymbol{x}^{\boldsymbol{X}-\boldsymbol{w}}$ is undefined.*

*Proof.* The proof is by induction on $m$. If $m = 0$ then the statement is trivial. For the induction step $m > 0$, we may assume that

$$\partial_{i_2} \cdots \partial_{i_m} \int \boldsymbol{x}^{\boldsymbol{X}} \, \mathrm{d}\mu(\boldsymbol{X}) = \int \partial_{i_2} \cdots \partial_{i_m} \boldsymbol{x}^{\boldsymbol{X}} \, \mathrm{d}\mu(\boldsymbol{X}) = \int p(\boldsymbol{X}) \boldsymbol{x}^{\boldsymbol{X}-\boldsymbol{w}} \, \mathrm{d}\mu(\boldsymbol{X})$$

By Lemma B.4, we reason

$$\partial_{i_1} \cdots \partial_{i_m} \int \boldsymbol{x}^{\boldsymbol{X}} \, \mathrm{d}\mu(\boldsymbol{X}) = \partial_{i_1} \int p(\boldsymbol{X}) \boldsymbol{x}^{\boldsymbol{X}-\boldsymbol{w}} \, \mathrm{d}\mu(\boldsymbol{X})$$
$$= \int (X_{i_1} - w_{i_1}) p(\boldsymbol{X}) \boldsymbol{x}^{\boldsymbol{X}-\boldsymbol{w}[i_1 \mapsto w_{i_1}+1]} \, \mathrm{d}\mu(\boldsymbol{X})$$
$$= \int \partial_{i_1} \cdots \partial_{i_m} \boldsymbol{x}^{\boldsymbol{X}} \, \mathrm{d}\mu(\boldsymbol{X})$$

which establishes the induction goal with the polynomial $\tilde{p}(\boldsymbol{X}) := (X_{i_1} - w_{i_1}) p(\boldsymbol{X})$ and $\tilde{\boldsymbol{w}} := \boldsymbol{w}[i_1 \mapsto w_{i_1} + 1]$. The property $\tilde{p}(\boldsymbol{X}) = 0$ whenever $X_j < \tilde{w}_j$ for some $j$ still holds due to the added factor $(X_{i_1} - w_{i_1})$ taking care of the case $X_{i_1} = w_{i_1} = \tilde{w}_{i_1} - 1$. $\qquad \square$

**Lemma B.4.** *Let $\boldsymbol{w} \in \mathbb{N}^n$, $p$ be a polynomial in $n$ variables, and $\mu$ a measure on $\mathbb{R}^n_{\geq 0}$. We define $p(\boldsymbol{X}) \boldsymbol{x}^{\boldsymbol{X}-\boldsymbol{w}} := 0$ whenever $p(\boldsymbol{X}) = 0$, even if $x_j \leq 0$ and $X_j < w_j$ for some $j$. Suppose $p(\boldsymbol{X}) \boldsymbol{x}^{\boldsymbol{X}-\boldsymbol{w}}$ is defined $\mu$-almost everywhere and $\mu$-integrable for all $\boldsymbol{x} \in T(R, \mu)$. Then*

$$\frac{\partial}{\partial x_i} \int p(\boldsymbol{X}) \boldsymbol{x}^{\boldsymbol{X}-\boldsymbol{w}} \, \mathrm{d}\mu(\boldsymbol{X}) = \int \frac{\partial}{\partial x_i} p(\boldsymbol{X}) \boldsymbol{x}^{\boldsymbol{X}-\boldsymbol{w}} \, \mathrm{d}\mu(\boldsymbol{X})$$
$$= \int (X_i - w_i) p(\boldsymbol{X}) \boldsymbol{x}^{\boldsymbol{X}-\boldsymbol{w}[i \mapsto w_i+1]} \, \mathrm{d}\mu(\boldsymbol{X})$$

*holds and is defined for all $\boldsymbol{x} \in T(R, \mu)$.*

*Proof.* The proof is about verifying the conditions of the Leibniz integral rule. The nontrivial condition is the boundedness of the derivative by an integrable function. The proof splits into two parts, depending on whether the $i$-th component of $\mu$ is supported on $\mathbb{N}$. If it is, then $x_i \in (-R, R)$ by the definition of $T(R, \mu)$ and can be nonpositive, but $X_i - w_i < 0$ and $p(\boldsymbol{X}) \neq 0$ happens $\mu$-almost never. Otherwise, we have $x \in (0, R)$ and it is thus guaranteed to be positive, but $X_i - w_i$ may be negative. Hence the two cases need slightly different treatment.

We first deal with the case that the $i$-th component is not supported on $\mathbb{N}$. Let $0 < \epsilon < R' < R'' < R$. We first prove $\frac{\partial}{\partial x_i} \int p(\boldsymbol{X}) \boldsymbol{x}^{\boldsymbol{X}} \, \mathrm{d}\mu(\boldsymbol{X}) = \int \frac{\partial}{\partial x_i} p(\boldsymbol{X}) \boldsymbol{x}^{\boldsymbol{X}} \, \mathrm{d}\mu(\boldsymbol{X})$ for any $x_i \in (\epsilon, R')$, using the Leibniz integral rule. For this purpose, we need to bound the derivative:

$$\left| \frac{\partial}{\partial x_i} p(\boldsymbol{X}) \boldsymbol{x}^{\boldsymbol{X}-\boldsymbol{w}} \right| \leq |(X_i - w_i) p(\boldsymbol{X})| \boldsymbol{x}^{\boldsymbol{X}-\boldsymbol{w}[i \mapsto w_i+1]} \leq \frac{1}{x_i} |X_i - w_i| |p(\boldsymbol{X})| \boldsymbol{x}^{\boldsymbol{X}-\boldsymbol{w}}$$
$$\leq \frac{1}{\epsilon} |X_i - w_i| |p(\boldsymbol{X})| \boldsymbol{x}^{\boldsymbol{X}-\boldsymbol{w}}$$

If $X_i > w_i$, we can choose $M > 1$ sufficiently large such that $|X_i - w_i| x_i^{X_i-w_i} \leq (X_i - w_i) R'^{X_i-w_i} \leq R''^{X_i-w_i}$ whenever $|X_i - w_i| > M$. (Such an $M$ can be found because the exponential function $y \mapsto \left( \frac{R''}{R'} \right)^y$ grows more quickly than the linear function $y \mapsto y$.) If $X_i - w_i \leq M$

then we also have $|X_i - w_i|x_i^{X_i - w_i} \leq M \cdot R''^{X_i - w_i}$. So the derivative is thus bounded on the set $\{\boldsymbol{X} \mid X_i > w_i\}$ by $\frac{M}{\epsilon} \cdot |p(\boldsymbol{X})|(\boldsymbol{x}[i \mapsto R''])^{\boldsymbol{X} - \boldsymbol{w}}$, which is integrable by assumption.

On the complement set $\{\boldsymbol{X} \mid 0 \leq X_i \leq w_i\}$, we find $|X_i - w_i|x_i^{X_i - w_i} \leq |X_i - w_i|\epsilon^{X_i - w_i} \leq w_i\epsilon^{X_i - w_i}$. Hence the derivative is bounded by $\frac{w_i}{\epsilon} \cdot |p(\boldsymbol{X})|(\boldsymbol{x}[i \mapsto \epsilon])^{\boldsymbol{X} - \boldsymbol{w}}$, which is again integrable by assumption. So the derivative is bounded by an integrable function and thus, by the Leibniz integral rule, interchanging differentiation and the integral is valid for all $x_i \in (\epsilon, R')$ and thus for all $x_i \in (0, R)$.

Next, consider the case that the $i$-th component of $\mu$ is supported on $\mathbb{N}$. Let $0 < R' < R'' < R$. Note that the set $\{\boldsymbol{X} \mid X_i < w_i \wedge p(\boldsymbol{X}) \neq 0\}$ has measure zero because otherwise the integrand $p(\boldsymbol{X})\boldsymbol{x}^{\boldsymbol{X} - \boldsymbol{w}}$ would not be defined for $x_i \leq 0$ due to the negative exponent $X_i - w_i$. Hence, for $\mu$-almost all $\boldsymbol{X}$, we have $p(\boldsymbol{X}) = [\boldsymbol{X} \geq \boldsymbol{w}]p(\boldsymbol{X})$ and can bound the partial derivative for $|x_i| < R'$:

$$
\begin{aligned}
\left|\frac{\partial}{\partial x_i}p(\boldsymbol{X})\boldsymbol{x}^{\boldsymbol{X} - \boldsymbol{w}}\right| &\leq |(X_i - w_i)p(\boldsymbol{X})||\boldsymbol{x}|^{\boldsymbol{X} - \boldsymbol{w}[i \mapsto w_i + 1]} \\
&\leq [\boldsymbol{X} \geq \boldsymbol{w}]|X_i - w_i||p(\boldsymbol{X})||\boldsymbol{x}|^{\boldsymbol{X} - \boldsymbol{w}[i \mapsto w_i + 1]} \\
&\leq [\boldsymbol{X} \geq \boldsymbol{w}[i \mapsto w_i + 1]]|X_i - w_i||p(\boldsymbol{X})||\boldsymbol{x}|^{\boldsymbol{X} - \boldsymbol{w}[i \mapsto w_i + 1]} \\
&\leq [\boldsymbol{X} \geq \boldsymbol{w}[i \mapsto w_i + 1]]|X_i - w_i||p(\boldsymbol{X})||\boldsymbol{x}[i \mapsto R']|^{\boldsymbol{X} - \boldsymbol{w}[i \mapsto w_i + 1]} \\
&\leq [\boldsymbol{X} \geq \boldsymbol{w}]|X_i - w_i||p(\boldsymbol{X})||\boldsymbol{x}[i \mapsto R']|^{\boldsymbol{X} - \boldsymbol{w}}
\end{aligned}
$$

because the factor $|X_i - w_i|$ vanishes for $X_i = w_i$ and thus ensures that negative values for the exponent $\boldsymbol{X} - \boldsymbol{w}[i \mapsto w_i + 1]$ don't matter, so that we can use the monotonicity of $|\boldsymbol{x}|^{\boldsymbol{X} - \boldsymbol{w}[i \mapsto w_i + 1]}$ in $|\boldsymbol{x}|$.

Similarly to the first case, for a sufficiently large $M > \max(1, w_i)$, we have $|X_i - w_i|R'^{X_i - w_i} \leq R''^{X_i - w_i}$ whenever $X_i - w_i > M$. If $X_i - w_i \leq M$ then $[\boldsymbol{X} \geq \boldsymbol{w}]|X_i - w_i| \leq M$ and we also have $[\boldsymbol{X} \geq \boldsymbol{w}]|X_i - w_i|R'^{X_i - w_i} \leq M \cdot R''^{X_i - w_i}$. The derivative is thus bounded by $M \cdot |p(\boldsymbol{X})||\boldsymbol{x}[i \mapsto R'']|^{\boldsymbol{X} - \boldsymbol{w}}$, which is integrable by assumption. By the Leibniz integral rule, interchanging differentiation and the integral is valid for all $x_i \in (-R', R')$ and thus for all $x_i \in (-R, R)$. $\qquad\square$

**Remark B.1.** As an example of what can go wrong with derivatives at zero if the measure is not supported on $\mathbb{N}$, consider the Dirac measure $\mu = \mathsf{Dirac}(\frac{1}{2})$. It has the generating function $G(x) := \mathsf{gf}(\mu)(x) = \sqrt{x}$ defined for $x \in \mathbb{R}_{\geq 0}$. Its derivative is $G'(x) := \frac{1}{2\sqrt{x}}$, which is not defined for $x = 0$.

### B.3 Possible extensions to the probabilistic programming language

The syntax of our language guarantees that the generating function of the distribution of any probabilistic program admits a closed form. But are the syntactic restrictions necessary or are there other useful programming constructs that can be supported? We believe the following constructs preserve closed forms of the generating function and could thus be supported:

- *Additional distributions with constant parameters:* any such distribution could be supported as long as its GF is defined on $[0, 1 + \epsilon)$ for some $\epsilon > 0$ and its derivatives can be evaluated.

- *Additional compound distributions:* we spent a lot of time trying to find additional compound distributions with a closed-form GF since this would be the most useful for probabilistic models, but we were largely unsuccessful. The only such distributions we found are $\mathsf{Binomial}(m, X_k)$, which could just be written as a sum of $m$ iid $\mathsf{Bernoulli}(X_k)$-variables, and $\mathsf{Gamma}(X_k, \beta)$ with shape parameter $X_k$, which could be translated to a GF with the same idea as for $\mathsf{Binomial}(X_k, p)$, $\mathsf{NegBinomial}(X_k, p)$, and $\mathsf{Poisson}(\lambda \cdot X_k)$.

- *Modulo, subtraction, iid sums:* The event $X_k \mod 2 = 0$, and the statements $X_k := \max(X_k - m, 0)$ (a form of subtraction that ensures nonnegativity), and $X_k := \mathsf{sum\_iid}(D, X_l)$ (i.e. $X_k$ is assigned the sum of $X_l$ independent random variables with distribution $D$) can also be supported as shown in [4, 18].

- *Nonlinear functions on variables with finite support*: if a variable has finite support, arbitrary functions on it could be performed by exhaustively testing all values in its domain.

- *Soft conditioning:* another supportable construct could be $\mathsf{score}\, a^{X_k}$ for $a \in [0,1]$, which multiplies the likelihood of the current path by $a^{X_k}$. We think $\mathsf{score}\, q(X_k)$ where $q(t) := \sum_{i=0}^m q_i t^i$ is a polynomial with coefficients $q_i \in [0,1]$ and with $q \leq 1$ on the domain of $X_k$ could also be supportable using similar techniques as for observations from $\mathsf{Bernoulli}(X_k)$.

None of these extensions seemed very useful for real-world models, in particular, they are not needed for our benchmarks. For this reason we did not formalize, implement, or prove them.

Support for loops is another big question. Bounded loops are easy enough to deal with by fully unrolling them. However exact inference for programs with unbounded loops seems extremely difficult, unless the inference algorithm is given additional information, e.g. a probabilistic loop invariant as in [4] or at least a loop invariant template with a few holes to be filled with real constants [18]. However, finding such a loop invariant for nontrivial programs seems exceedingly difficult. Furthermore, the variables of looping programs may have infinite expected values, so the generating function may not be definable at the point 1. But in our semantics, evaluating at 1 is very important for marginalization. For these reasons, we consider exact inference in the presence of loops an interesting but very hard research problem.

## C   Details on implementation and optimizations

**Overview**   Our tool takes an SGCL program $P$ with $n$ variables as input, and a variable $X_i$ whose posterior distribution is to be computed. It translates it to the GF $[\![P]\!]_{\mathsf{gf}}(\mathbf{1})$ according to the GF semantics and normalizes it: $G := \mathsf{normalize}([\![P]\!]_{\mathsf{gf}}(\mathbf{1}))$. This $G$ is the GF of the posterior distribution of the program, from which it extracts the posterior moments of $X_i$ and its probability masses (if $X_i$ is discrete).

**Computation of moments**   First, it computes the first four (raw) posterior moments from the factorial moments, which are obtained by differentiation of $G$:

$$M_1 := \mathbb{E}[X_i] = \partial_i G(\mathbf{1}_n)$$
$$M_2 := \mathbb{E}[X_i^2] = \mathbb{E}[(X_i)(X_i - 1)] + \mathbb{E}[X_i] = \partial_i^2 G(\mathbf{1}_n) + \partial_i G(\mathbf{1}_n)$$
$$M_3 := \mathbb{E}[X_i^3] = \mathbb{E}[X_i(X_i - 1)(X_i - 2)] + 3\mathbb{E}[X_i(X_i - 1)] + \mathbb{E}[X_i]$$
$$= \partial_i^3 G(\mathbf{1}_n) + 3\partial_i^2 G(\mathbf{1}_n) + \partial_i G(\mathbf{1}_n)$$
$$M_4 := \mathbb{E}[X_i^4] = \mathbb{E}[X_i(X_i - 1)(X_i - 2)(X_i - 3)] + 6\mathbb{E}[X_i(X_i - 1)(X_i - 2)]$$
$$+ 7\mathbb{E}[X_i(X_i - 1)] + \mathbb{E}[X_i]$$
$$= \partial_i^4 G(\mathbf{1}_n) + 6\partial_i^3 G(\mathbf{1}_n) + 7\partial_i^2 G(\mathbf{1}_n) + \partial_i G(\mathbf{1}_n)$$

From the raw moments, it computes the first four (centered/standardized) moments, i.e. the expected value ($\mu := \mathbb{E}[X_i]$), the variance ($\sigma^2 := \mathbb{V}[X_i]$), the skewness ($\mathbb{E}[(X_i - \mu)^3]/\sigma^3$) and the kurtosis ($\mathbb{E}[(X_i - \mu)^4]/\sigma^4$):

$$\mu := \mathbb{E}[X_i] = M_1$$
$$\sigma^2 := \mathbb{V}[X_i] = \mathbb{E}[(X_i - \mu)^2] = \mathbb{E}[X_i^2] - 2\mu\mathbb{E}[X_i] + \mu^2 = M_2 - \mu^2$$
$$\mathsf{Skew}[X_i] = \frac{\mathbb{E}[(X_i - \mu)^3]}{\sigma^3} = \frac{\mathbb{E}[X_i^3] - 3\mu\mathbb{E}[X_i^2] + 3\mu^2\mathbb{E}[X_i] - \mu^3}{\sigma^3}$$
$$= \frac{M_3 - 3\mu M_2 + 2\mu^3}{\sigma^3}$$
$$\mathsf{Kurt}[X_i] = \frac{\mathbb{E}[(X_i - \mu)^4]}{\sigma^4} = \frac{\mathbb{E}[X_i^4] - 4\mu\mathbb{E}[X_i^3] + 6\mu^2\mathbb{E}[X_i^2] - 4\mu^3\mathbb{E}[X_i] + \mu^4}{\sigma^4}$$
$$= \frac{M_4 - 4\mu M_3 + 6\mu^2 M_2 - 3\mu^4}{\sigma^4}$$

**Computation of probability masses**   If $X_i$ is a discrete variable, the tool computes all the probability masses $\mathbb{P}[X_i = m]$ until the value of $m$ such that the tail probabilities are guaranteed to be below a certain threshold, which is set to $\mathbb{P}[X_i \geq m] \leq \frac{1}{256}$. This is achieved by setting

$m := \mu + 4\sqrt[4]{\mathbb{E}[(X_i - \mu)^4]}$ where $\mu := \mathbb{E}[X_i]$. Then we find

$$\mathbb{P}[X_i \geq m] \leq \mathbb{P}\left[|X_i - \mu| \geq 4\sqrt[4]{\mathbb{E}[(X_i - \mu)^4]}\right] = \mathbb{P}\left[(X - \mu)^4 \geq 256\mathbb{E}[(X_i - \mu)^4]\right] \leq \frac{1}{256}$$

by Markov's inequality. Note that in practice, the tail probabilities are typically much smaller than $\frac{1}{256}$, usually in the order of $10^{-5}$. For all $k \in \{0, \ldots, m\}$, the tool computes the posterior probability $\mathbb{P}[X_i = k]$ by marginalizing out all the other variables (substituting 1 for them) and computing the Taylor coefficient at $x_i = 0$:

$$\mathbb{P}[X_i = k] = \frac{1}{k!}\partial_i^k G(\mathbf{1}_n[i \mapsto 0]).$$

It would be desirable to compute posterior *densities* for continuous distributions as well. In fact, there are mathematical ways of recovering the probability density function from a generating function via an inverse Laplace transform. However, this cannot be automated in practice because it requires solving integrals, which is intractable.

**Implementation details**   Our tool Genfer is implemented in Rust [20], a safe systems programming language. The main reasons were low-level control and performance: the operations on the Taylor polynomials to evaluate derivatives of the generating function need to be fast and are optimized to exploit the structure of GFs arising from probabilistic programs (see Appendix C.3). C or C++ would have satisfied the performance criterion as well, but Rust's language features like memory safety, enums (tagged unions), and pattern matching made the implementation a lot more pleasant, robust, and easier to maintain. The first author's experience with Rust was another contributing factor.

The coefficients of the Taylor polynomials are stored in a multidimensional array provided by the `ndarray` library[4]. Arbitrary-precision floating-point numbers and unbounded rational numbers are provided by the `rug` library[5], which is an interface to the GNU libraries GMP (for rationals) and MPFR (for floats). Our implementation is available on GitHub: github.com/fzaiser/genfer

### C.1   Automatic differentiation and Taylor polynomials

The main difficulty in implementing the GF semantics is the computation of the (partial) derivatives. This seems to be a unique situation where we want to compute $d$-th derivatives where $d$ is in the order of hundreds. The reason is that the observe $X_k = d$ construct is translated into a $d$-th (partial) derivative and if $d$ is a real-world observation, it can be large. We have not come across another application of automatic differentiation that required derivatives of a total order of more than 10.

As a consequence, when we tried *PyTorch*, a mature machine learning library implementing automatic differentiation, the performance for derivatives of high order was very poor. Therefore, we decided to implement our own automatic differentiation framework. Our approach is to compute the Taylor expansion in all variables up to some order $d$ of the generating function instead of a symbolic expression of the $d$-th derivative. This has the advantage of growing polynomially in $d$, not exponentially. Contrary to [27], it is advantageous to use Taylor coefficients instead of the partial derivatives because the additional factorial factors are can easily lead to overflows.

**Taylor polynomials**   More formally, we define the *Taylor polynomial* $\mathsf{Taylor}_{\boldsymbol{w}}^d(G)$ of a function $G : \mathbb{R}^n \to \mathbb{R}$ at $\boldsymbol{w} \in \mathbb{R}^n$ of order $d$ as the polynomial

$$\mathsf{Taylor}_{\boldsymbol{w}}^d(G) := \sum_{\boldsymbol{\alpha} \in \mathbb{N}^n : |\boldsymbol{\alpha}| \leq d} \frac{1}{\boldsymbol{\alpha}!}\partial^{\boldsymbol{\alpha}} G(\boldsymbol{w}) \cdot (\boldsymbol{x} - \boldsymbol{w})^{\boldsymbol{\alpha}}$$

where we used multi-index notation: $|\boldsymbol{\alpha}|$ stands for $\alpha_1 + \cdots + \alpha_n$; $\boldsymbol{\alpha}!$ for $\alpha_1! \cdots \alpha_n!$; and $\partial^{\boldsymbol{\alpha}}$ for $\partial_1^{\alpha_1} \ldots \partial_n^{\alpha_n}$. The coefficients $c_{\boldsymbol{\alpha}} := \frac{1}{\boldsymbol{\alpha}!}\partial^{\boldsymbol{\alpha}} G(\boldsymbol{w})$ are the *Taylor coefficients* of $G$ at $\boldsymbol{w}$ and are stored in a multidimensional array in our implementation.

---

[4] https://crates.io/crates/ndarray
[5] https://crates.io/crates/rug

**Operations** The operations on generating functions (Appendix B) are implemented in terms of their effect on the Taylor expansions. Let $G, H : \mathbb{R}^n \to \mathbb{R}$ be two functions with Taylor expansions $\mathsf{Taylor}^d_{\boldsymbol{w}}(G) = \sum_{|\boldsymbol{\alpha}| \leq d} g_{\boldsymbol{\alpha}}(\boldsymbol{x} - \boldsymbol{w})^{\boldsymbol{\alpha}}$ and $\mathsf{Taylor}^d_{\boldsymbol{w}}(H) = \sum_{|\boldsymbol{\alpha}| \leq d} h_{\boldsymbol{\alpha}}(\boldsymbol{x} - \boldsymbol{w})^{\boldsymbol{\alpha}}$. Addition $F = G + H$ of two generating functions is implemented by adding the coefficients: $\mathsf{Taylor}^d_{\boldsymbol{w}}(G + H) = \sum_{|\boldsymbol{\alpha}| \leq d}(g_{\boldsymbol{\alpha}} + h_{\boldsymbol{\alpha}})(\boldsymbol{x} - \boldsymbol{w})^{\boldsymbol{\alpha}}$. Scalar multiplication $F = c \cdot G$ is implemented by multiplying the coefficients: $\mathsf{Taylor}^d_{\boldsymbol{w}}(c \cdot G) = \sum_{|\boldsymbol{\alpha}| \leq d}(c \cdot g_{\boldsymbol{\alpha}})(\boldsymbol{x} - \boldsymbol{w})^{\boldsymbol{\alpha}}$. Multiplication $F = G \cdot H$ of two generating functions is implemented by the Cauchy product:

$$\mathsf{Taylor}^d_{\boldsymbol{w}}(G \cdot H) = \sum_{|\boldsymbol{\alpha}| \leq d} \left( \sum_{\boldsymbol{\alpha}_1 + \boldsymbol{\alpha}_2 = \boldsymbol{\alpha}} g_{\boldsymbol{\alpha}_1} h_{\boldsymbol{\alpha}_2} \right) (\boldsymbol{x} - \boldsymbol{w})^{\boldsymbol{\alpha}}.$$

Division, exponentiation, logarithms, and powers can be implemented as well (see [13, Chapter 13] for details). Partial derivatives essentially correspond to shifting the index and a multiplication:

$$\mathsf{Taylor}^{d-1}_{\boldsymbol{w}}(\partial_k G) = \sum_{|\boldsymbol{\alpha}| \leq d-1} (\alpha_k + 1) g_{\boldsymbol{\alpha}[k \mapsto \alpha_k + 1]} \boldsymbol{x}^{\boldsymbol{\alpha}}$$

The most complicated case is composition/substitution, i.e. $F(\boldsymbol{x}) = G(\boldsymbol{x}[k \mapsto H(\boldsymbol{x})])$. For this, we let $\boldsymbol{w}' := w[k \mapsto h_{\boldsymbol{0}}]$ where $\mathsf{Taylor}^d_{\boldsymbol{w}}(H) = \sum_{|\boldsymbol{\beta}| \leq d} h_{\boldsymbol{\beta}}(\boldsymbol{x} - \boldsymbol{w})^{\boldsymbol{\beta}}$ and we let $\mathsf{Taylor}^d_{\boldsymbol{w}'}(G) = \sum_{|\boldsymbol{\alpha}| \leq d} g_{\boldsymbol{\alpha}}(\boldsymbol{x} - \boldsymbol{w}')^{\boldsymbol{\alpha}}$. Then we have (where "h.o.t." stands for "higher order terms")

$$G(\boldsymbol{x}[k \mapsto H(\boldsymbol{x})]) = G\left( \boldsymbol{x}\left[ k \mapsto \sum_{|\boldsymbol{\beta}| \leq d} h_{\boldsymbol{\beta}}(\boldsymbol{x} - \boldsymbol{w})^{\boldsymbol{\beta}} + \text{h.o.t.} \right] \right)$$

$$= \sum_{|\boldsymbol{\alpha}| \leq d} g_{\boldsymbol{\alpha}} \left( \boldsymbol{x}\left[ k \mapsto h_{\boldsymbol{0}} + \sum_{1 \leq |\boldsymbol{\beta}| \leq d} h_{\boldsymbol{\alpha}}(\boldsymbol{x} - \boldsymbol{w})^{\boldsymbol{\beta}} + \text{h.o.t.} \right] - \boldsymbol{w}' \right)^{\boldsymbol{\alpha}} + \text{h.o.t.}$$

$$= \sum_{|\boldsymbol{\alpha}| \leq d} g_{\boldsymbol{\alpha}} \left( (\boldsymbol{x} - \boldsymbol{w})\left[ k \mapsto \sum_{1 \leq |\boldsymbol{\beta}| \leq d} h_{\boldsymbol{\alpha}}(\boldsymbol{x} - \boldsymbol{w})^{\boldsymbol{\beta}} \right] \right)^{\boldsymbol{\alpha}} + \text{h.o.t.}$$

because by definition $\boldsymbol{x}[k \mapsto h_{\boldsymbol{0}}] - w' = (\boldsymbol{x} - \boldsymbol{w})[k \mapsto 0]$. This means that we can obtain $\mathsf{Taylor}^d_{\boldsymbol{w}}(F)$ by substituting $\sum_{1 \leq |\boldsymbol{\beta}| \leq d} h_{\boldsymbol{\alpha}}(\boldsymbol{x} - \boldsymbol{w})^{\boldsymbol{\beta}}$ for $x_k - w'_k$ in $\mathsf{Taylor}^d_{\boldsymbol{w}'}(G)$.

**Example C.1** (Calculations with Taylor polynomials). Here we show how the expectation for Example 2.1 can be computed using Taylor polynomials. We use the same naming as in Example 3.1. Recall that to compute $\mathbb{E}[X]$, we need to evaluate $\partial_1 E(1, 1)$, so we compute the Taylor expansion of $E$ at $(1, 1)$ of order 1, i.e. $\mathsf{Taylor}^1_{(1,1)}(E)$. To compute $E(x, y) = \frac{D(x,y)}{D(1,1)}$, we need $D(x, y)$, so we need $\mathsf{Taylor}^1_{(1,1)}(D)$. For $D(x, y) = \frac{1}{2!} y^2 \partial_y^2 C(x, 0)$, we need the Taylor expansion of $C$ at $(1, 0)$ of order $1 + 2 = 3$. For $C(x, y) = B(x(0.1y + 0.9), 1)$, we need the Taylor expansion of $B$ at $(1(0.1 \cdot 0 + 0.9), 1) = (0.9, 1)$ of order 3. Since $B(x, y) = \exp(20(x - 1)) = \exp(20((x - 0.9) - 0.1)) = \exp(20(x - 0.9) - 2)$, this Taylor expansion is

$$\mathsf{Taylor}^3_{(0.9,1)}(B) = e^{-2} + 20e^{-2} \cdot (x - 0.9) + 200e^{-2} \cdot (x - 0.9)^2 + \frac{4000}{3} e^{-2}(x - 0.9)^3.$$

Since $C(1 + (x - 1), 0 + y) = B((1 + (x - 1))(0.1y + 0.9), 1) = B(0.9 + 0.9(x - 1) + 0.1y + 0.1(x - 1)y, 1)$, the Taylor expansion of $C$ at $(1, 0)$ is given by replacing $(x - 0.9)$ with $0.9(x - 1) + 0.1y + 0.1(x - 1)y$ and we get

$$\mathsf{Taylor}^3_{(1,0)}(C) = e^{-2}\big(1 + 2y + 2y^2 + \frac{4}{3}y^3 + 18(x - 1) + 38(x - 1)y + 40(x - 1)y^2$$
$$+ 162(x - 1)^2 + 360(x - 1)^2 y + 972(x - 1)^3\big)$$

For $D(x, y) = \frac{1}{2!} y^2 \partial_y^2 C(x, 0)$, we compute the second partial derivative wrt. $y$, substitute $y = 0$ and then multiply by $y^2 = (1 + (y - 1))^2$, yielding

$$D(x, y) = \frac{1}{2!} y^2 e^{-2} \left. \left(4 + 8y + 80(x - 1) + \text{higher order terms}\right)\right|_{y=0}$$
$$= e^{-2}(4 + 80(x - 1) + \text{higher order terms})(1 + (y - 1))^2$$
$$= e^{-2}(4 + 80(x - 1) + 8(y - 1) + \text{higher order terms})$$

As a consequence, we find that $D(1, 1) = 4e^{-2}$, so $\mathsf{Taylor}^1_{(1,1)}(E) = 1 + 20(x - 1) + 2(y - 1)$. So $\mathbb{E}[X] = \partial_x E(1, 1) = 20$, as desired.

**Memoizing intermediate results** When computing the Taylor expansion of a function $G$, it is important to memoize the intermediate Taylor expansions of subexpressions of $G$ if they occur more than once. For example, conditionals are translated to a generating function that uses the previous generating function twice. Evaluating it repeatedly would double the running time for each conditional. With memoization, it is possible to handle programs with lots of branching (e.g. $> 2^{100}$ paths in the mixture model, cf. Table 5) in a reasonable amount of time.

## C.2    Optimizing Observations from Compound Distributions

Observations from compound distributions are generally the bottleneck for the running time because the construct observe $d \sim D(X_k)$ is syntactic sugar for $X_{n+1} \sim D(X_k)$; observe $X_{n+1} = d$, which introduces a new variable, which is immediately discarded after the observation. This expansion has the semantics

$$[\![\text{observe } d \sim D(X_k)]\!]_{\mathsf{gf}}(G)(\boldsymbol{x}) = [\![\text{observe } X_{n+1} = d]\!]_{\mathsf{gf}}([\![X_{n+1} \sim D(X_k)]\!]_{\mathsf{gf}}(G))(x_1, \ldots, x_{n+1})$$

$$= \frac{1}{d!} \frac{\partial^d}{\partial x_{n+1}^d} [\![X_{n+1} \sim D(X_k)]\!]_{\mathsf{gf}}(G)(x_1, \ldots, x_{n+1}) \Big|_{x_{n+1}=0}$$

containing an extra variable $x_{n+1}$, which worsens the running time of the Taylor approach significantly. We can find more efficient ways of computing this semantics.

**Theorem C.1.** *Observing from compound binomial distributions can be implemented without introducing a new variable:*

$$[\![\text{observe } d \sim \mathsf{Binomial}(X_k, p)]\!]_{\mathsf{gf}}(G) = \frac{1}{d!}(px_k)^d \cdot \partial_k^d G(\boldsymbol{x}[k \mapsto (1-p)x_k])$$

*Proof.* A proof for the discrete setting was given in [27]. In our setting, which allows continuous distributions, we cannot use the usual argument about power series anymore. Instead, we reason as follows:

$$[\![\text{observe } d \sim \mathsf{Binomial}(X_k, p)]\!]_{\mathsf{gf}}(G)$$

$$= \frac{1}{d!} \frac{\partial^d}{\partial x_{n+1}^d} [\![X_{n+1} \sim \mathsf{Binomial}(X_k)]\!]_{\mathsf{gf}}(G)(x_1, \ldots, x_{n+1}) \Big|_{x_{n+1}=0}$$

$$= \frac{1}{d!} \frac{\partial^d}{\partial x_{n+1}^d} G(\boldsymbol{x}[k \mapsto x_k \cdot (px_{n+1} + 1 - p)]) \Big|_{x_{n+1}=0}$$

$$\overset{*}{=} \frac{1}{d!} \partial_k^d G(\boldsymbol{x}[k \mapsto x_k \cdot (px_{n+1} + 1 - p)]) \cdot (px_k)^d \Big|_{x_{n+1}=0}$$

$$= \frac{1}{d!}(px_k)^d \partial_k^d G(\boldsymbol{x}[k \mapsto (1-p)x_k])$$

where $*$ follows by iterating the following argument:

$$\frac{\partial}{\partial x_{n+1}} G(\boldsymbol{x}[k \mapsto x_k \cdot (px_{n+1} + 1 - p)])$$

$$= \partial_k G(\boldsymbol{x}[k \mapsto x_k \cdot (px_{n+1} + 1 - p)]) \cdot \frac{\partial(x_k \cdot (px_{n+1} + 1 - p))}{\partial x_{n+1}}$$

$$= \partial_k G(\boldsymbol{x}[k \mapsto x_k \cdot (px_{n+1} + 1 - p)]) \cdot px_k$$

which holds by the chain rule. □

**Theorem C.2.** *Observing from compound Poisson distributions can be implemented without introducing a new variable:*

$$[\![\text{observe } d \sim \mathsf{Poisson}(\lambda X_k)]\!]_{\mathsf{gf}}(G) = \frac{1}{d!} D_{k,\lambda}^d(G)(\boldsymbol{x}[k \mapsto e^{-\lambda}x_k])$$

*where*

$$D_{k,\lambda}(G)(\boldsymbol{x}) := \lambda x_k \cdot \partial_k G(\boldsymbol{x}).$$

*Proof.* Let $F : \mathbb{R}^n \to \mathbb{R}$ be a smooth function. Then

$$\frac{\partial}{\partial x_{n+1}} F(\boldsymbol{x}[k \mapsto x_k \cdot e^{\lambda(x_{n+1}-1)}])$$

$$= \partial_k F(\boldsymbol{x}[k \mapsto x_k \cdot e^{\lambda(x_{n+1}-1)}]) \cdot x_k e^{\lambda(x_{n+1}-1)} \cdot \lambda$$

$$= D_{k,\lambda}(F)(\boldsymbol{x}[k \mapsto x_k \cdot e^{\lambda(x_{n+1}-1)}]).$$

Inductively, we get

$$\frac{\partial^d}{\partial x_{n+1}^d} G(\boldsymbol{x}[k \mapsto x_k \cdot e^{\lambda(x_{n+1}-1)}]) = D_{k,\lambda}^d(G)(\boldsymbol{x}[k \mapsto x_k \cdot e^{\lambda(x_{n+1}-1)}]).$$

Substituting $x_{n+1} \mapsto 0$ in the final expression and dividing by $d!$ yields:

$$[\![\text{observe } d \sim \mathsf{Poisson}(\lambda X_k)]\!]_{\mathsf{gf}}(G)$$

$$= \frac{1}{d!} \frac{\partial^d}{\partial x_{n+1}^d} [\![X_{n+1} \sim \mathsf{Poisson}(\lambda X_k)]\!]_{\mathsf{gf}}(G)(x_1, \dots, x_{n+1}) \Big|_{x_{n+1}=0}$$

$$= \frac{1}{d!} \frac{\partial^d}{\partial x_{n+1}^d} G(\boldsymbol{x}[k \mapsto x_k \cdot e^{\lambda(x_{n+1}-1)}]) \Big|_{x_{n+1}=0}$$

$$= \frac{1}{d!} D_{k,\lambda}^d(G)(\boldsymbol{x}[k \mapsto e^{-\lambda} x_k])$$

$\square$

**Theorem C.3.** *Observing from compound negative binomial distributions can be implemented without introducing a new variable:*

$$[\![\text{observe } d \sim \mathsf{NegBinomial}(X_k, p)]\!]_{\mathsf{gf}}(G) = \frac{1}{d!} \sum_{i=0}^{d} \partial_k^i F(\boldsymbol{x}[k \mapsto p \cdot x_k]) \cdot p^i x_k^i \cdot (1-p)^d L_{d,i}$$

*where the numbers $L_{d,i}$ are known as Lah numbers and defined recursively as follows:*

$$L_{0,0} := 1$$
$$L_{d,i} := 0 \qquad\qquad\qquad \text{for } i < 0 \text{ or } d < i$$
$$L_{d+1,i} := (d+i)L_{d,i} + L_{d,i-1} \qquad\qquad \text{for } 0 \leq i \leq d+1$$

*Proof.* Let $F : \mathbb{R}^n \to \mathbb{R}$ be a smooth function and $y : \mathbb{R} \to \mathbb{R}$ given by $y(z) := \frac{1}{1-(1-p)z}$. Then $y'(x) = -\frac{1}{(1-(1-p)x)^2} \cdot (-(1-p)) = (1-p)y(x)^2$ and

$$\frac{\partial}{\partial x_{n+1}} F\left(\boldsymbol{x}\left[k \mapsto \frac{x_k \cdot p}{1 - (1-p)x_{n+1}}\right]\right)$$

$$= \partial_k F(\boldsymbol{x}[k \mapsto x_k p \cdot y(x_{n+1})]) \cdot p x_k \cdot (1-p)y(x_{n+1})^2$$

We claim that we can write

$$\frac{\partial^d}{\partial x_{n+1}^d} F\left(\boldsymbol{x}\left[k \mapsto \frac{p x_k}{1 - (1-p)x_{n+1}}\right]\right)$$

$$= \sum_{i=0}^{d} \partial_k^i F(\boldsymbol{x}[k \mapsto x_k \cdot y(x_{n+1})]) \cdot p^i x_k^i \cdot (1-p)^d L_{d,i} \cdot y(x_{n+1})^{d+i}.$$

The claim is proved by induction. First the case $d = 0$:

$$F\left(\boldsymbol{x}\left[k \mapsto \frac{p x_k}{1 - (1-p)x_{n+1}}\right]\right) = F\left(\boldsymbol{x}\left[k \mapsto \frac{p x_k}{1 - (1-p)x_{n+1}}\right]\right) \cdot (1-p)^0 x_k^0 L_{0,0} \cdot y(x_{n+1})^0$$

Next, the induction step:

$$\frac{\partial^{d+1}}{\partial x_{n+1}^{d+1}} F\left(\boldsymbol{x}\left[k \mapsto \frac{p x_k}{1 - (1-p)x_{n+1}}\right]\right)$$

$$= \frac{\partial}{\partial x_{n+1}} \left( \sum_{i=0}^{d} \partial_k^i F(\boldsymbol{x}[k \mapsto px_k \cdot y(x_{n+1})]) \cdot p^i x_k^i \cdot (1-p)^d L_{d,i} \cdot y(x_{n+1})^{d+i} \right)$$

$$= \sum_{i=0}^{d} \partial_k^{i+1} F(\boldsymbol{x}[k \mapsto px_k \cdot y(x_{n+1})]) \cdot px_k y'(x_{n+1}) \cdot p^i x_k^i (1-p)^d L_{d,i} y(x_{n+1})^{d+i}$$

$$+ \sum_{i=0}^{d} \partial_k^i F(\boldsymbol{x}[k \mapsto px_k \cdot y(x_{n+1})]) \cdot p^i x_k^i (1-p)^d \cdot L_{d,i}(d+i) y(x_{n+1})^{d+i-1} \cdot y'(x_{n+1})$$

$$= \sum_{i=0}^{d} \partial_k^{i+1} F(\boldsymbol{x}[k \mapsto px_k \cdot y(x_{n+1})]) \cdot p^{i+1} x_k^{i+1} \cdot (1-p)^{d+1} y(x_{n+1})^2 \cdot L_{d,i} y(x_{n+1})^{d+i}$$

$$+ \sum_{i=0}^{d} \partial_k^i F(\boldsymbol{x}[k \mapsto px_k \cdot y(x_{n+1})]) \cdot p^i x_k^i \cdot (1-p)^{d+1} y(x_{n+1})^2 \cdot (d+i) L_{d,i} y(x_{n+1})^{d+i-1}$$

$$= \sum_{i=0}^{d+1} \partial_k^i F(\boldsymbol{x}[k \mapsto px_k \cdot y(x_{n+1})]) \cdot p^i x_k^i \cdot (1-p)^{d+1} L_{d,i-1} y(x_{n+1})^{d+i+1}$$

$$+ \sum_{i=0}^{d+1} \partial_k^i F(\boldsymbol{x}[k \mapsto px_k \cdot y(x_{n+1})]) \cdot p^i x_k^i \cdot (1-p)^{d+1} (d+i) L_{d,i} y(x_{n+1})^{d+i+1}$$

$$= \sum_{i=0}^{d+1} \partial_k^i F(\boldsymbol{x}[k \mapsto px_k \cdot y(x_{n+1})]) \cdot p^i x_k^i \cdot (1-p)^{d+1} ((d+i) L_{d,i} + L_{d,i-1}) y(x_{n+1})^{d+i+1}$$

$$= \sum_{i=0}^{d+1} \partial_k^i F(\boldsymbol{x}[k \mapsto px_k \cdot y(x_{n+1})]) \cdot p^i x_k^i \cdot (1-p)^{d+1} L_{d+1,i} y(x_{n+1})^{d+i+1}$$

Substituting $x_{n+1} \mapsto 0$ in the final expression and dividing by $d!$ yields:

$$[\![ \text{observe}\, d \sim \mathsf{NegBinomial}(X_k, p) ]\!]_{\mathsf{gf}}(G)$$

$$= \frac{1}{d!} \frac{\partial^d}{\partial x_{n+1}^d} [\![ X_{n+1} \sim \mathsf{NegBinomial}(X_k, p) ]\!]_{\mathsf{gf}}(G)(x_1, \ldots, x_{n+1}) \Big|_{x_{n+1}=0}$$

$$= \frac{1}{d!} \frac{\partial^d}{\partial x_{n+1}^d} G(\boldsymbol{x}[k \mapsto \frac{x_k \cdot p}{1 - (1-p)x_{n+1}}]) \Big|_{x_{n+1}=0}$$

$$= \frac{1}{d!} \sum_{i=0}^{d} \partial_k^i F(\boldsymbol{x}[k \mapsto p \cdot x_k]) \cdot p^i x_k^i \cdot (1-p)^d L_{d,i}$$

$\square$

**Theorem C.4.** *Observing from compound Bernoulli distributions can be implemented without introducing a new variable:*

$$[\![ \text{observe}\, d \sim \mathsf{Bernoulli}(X_k) ]\!]_{\mathsf{gf}}(G) = \begin{cases} G(\boldsymbol{x}) - x_k \partial_j G(\boldsymbol{x}) & d = 0 \\ x_k \cdot \partial_j G(\boldsymbol{x}) & d = 1 \\ 0 & \text{otherwise} \end{cases}$$

*Proof.* We argue as follows:

$$[\![ \text{observe}\, d \sim \mathsf{Bernoulli}(X_k) ]\!]_{\mathsf{gf}}(G)(\boldsymbol{x})$$

$$= \frac{1}{d!} \frac{\partial^d}{\partial x_{n+1}^d} [\![ X_{n+1} \sim \mathsf{Bernoulli}(X_k) ]\!]_{\mathsf{gf}}(G)(x_1, \ldots, x_{n+1}) \Big|_{x_{n+1}=0}$$

$$= \frac{1}{d!} \frac{\partial^d}{\partial x_{n+1}^d} (G(\boldsymbol{x}) + x_k(x_{n+1} - 1) \cdot \partial_k G(\boldsymbol{x})) \Big|_{x_{n+1}=0}$$

If $d = 0$, the right-hand side is $G(\boldsymbol{x}) - x_k \partial_k G(\boldsymbol{x})$. If $d = 1$, the right-hand side is $x_k \partial_k G(\boldsymbol{x})$. For larger $d \geq 2$, the $d$-th derivative vanishes, which completes the proof. $\square$

### C.3 Analysis of the running time

If the program makes observations (or compares variables with constants) $d_1, \ldots, d_m$, the running time of the program will depend on these constants. Indeed, each observation of (or comparison of a variable with) $d_i$ requires the computation of the $d_i$-th partial derivative of the generating function. In our Taylor polynomial approach, this means that the highest order of Taylor polynomials computed will be $d := \sum_{i=1}^m d_i$. If the program has $n$ variables, storing the coefficients of this polynomial alone requires $O(d^n)$ space. This can be reduced if the generating function is a low-degree polynomial, in which case we do not store the zero coefficients of the terms of higher order.

Naively, the time complexity of multiplying two such Taylor polynomials is $O(d^{2n})$ and that of composing them is $O(d^{3n})$. However, we can do better by exploiting the fact that the polynomials are sparse. In the generating function semantics, we never multiply two polynomials with $n$ variables. One of them will have at most two variables. This brings the running time for multiplications down to $O(d^{n+2})$. Furthermore, the GF semantics has the property that when composing two generating functions, we can substitute variables one by one instead of simultaneously, bringing the running time down to $O(n \cdot d^{2n+1})$. Even more, at most one variable is substituted and the substituted terms have at most two variables, which brings the running time down to $O(d^{n+3})$.

**Theorem C.5.** *Our exact inference method can evaluate the $k$-th derivative of a generating function in $O(s \cdot d^{n+3})$, and $O(sd^3)$ for $n = 1$, where $s$ is the number of statements, $d$ (for data) is $k$ plus the sum of all values that are observed or compared against in the program, and $n$ is the number of variables of the program.*

*Proof.* The above arguments already analyze the bottlenecks of the algorithm. But for completeness' sake, we consider all operations being performed on generating functions and their Taylor expansions (cf. Appendix C.1).

The maximum order of differentiation needed is $d$ and since there are $n$ variables, the maximum number of Taylor coefficients that need to be stored is $O(d^n)$. Addition of two such Taylor polynomials and scalar multiplications trivially take $O(d^n)$ time. Since in the GF semantics, a GF is only ever multiplied by one with at most two variables, the multiplications take $O(d^{n+2})$ time. Division only happens with a divisor of degree at most 1, which can also be implemented in $O(d^{n+1})$ time [13, Chapter 13]. Exponentiation $\exp(\dots)$, logarithms $\log(\dots)$, and powers $(\dots)^m$ are only performed on polynomials with one variable, where they can be implemented in $O(d^2)$ time [13, Table 13.2]. Finally, the only substitutions that are required are ones where a term with at most two variables is substituted for a variable, i.e. of the form $p(\boldsymbol{x}[k \mapsto q(x_1, x_2)])$ where $p(\boldsymbol{x}) = \sum_{|\boldsymbol{\alpha}| \leq d} p_{\boldsymbol{\alpha}} \boldsymbol{x}^{\boldsymbol{\alpha}}$. Then

$$p(\boldsymbol{x}[k \mapsto q(x_1, x_2)]) = p_0(\boldsymbol{x}) + q(x_1, x_2) \cdot (p_1(\boldsymbol{x}) + q(x_1, x_2) \cdot (p_2(\boldsymbol{x}) + \cdots))$$

where $p_i(\boldsymbol{x}) = \sum_{\boldsymbol{\alpha} : \alpha_k = i} p_{\boldsymbol{\alpha}} \boldsymbol{x}^{\boldsymbol{\alpha}[k \mapsto 0]}$. This needs $d$ multiplications of $q(x_1, x_2)$ in 2 variables and a polynomial in $n$ variables, each of which takes $O(d^{n+2})$ time. In total, the substitution can be performed in $O(d^{n+3})$ time. Finally, if there is only one variable, the substitution can be performed in $O(d^3)$ time because $q$ can only have one variable.

In summary, the input SGCL program has $s$ statements, and each of them is translated to a bounded number of operations on generating functions. These operations are performed using Taylor expansions of degree at most $d$, each of which takes at most $O(d^{n+3})$ time (and at most $O(d^3)$ time for $n = 1$). So overall, the running time is $O(s \cdot d^{n+3})$ (and $O(sd^3)$ for $n = 1$). $\square$

**Potential improvements of the running time** Using various tricks, such as FFT, the running time of the composition of Taylor polynomials, i.e. substitution of polynomials, can be improved to $O((d \log d)^{3n/2})$ [2]. This is still exponential in the number of variables $n$, but would improve our asymptotic running time for $n < 6$ program variables. In fact, we experimented with an implementation of multiplication and composition based on the fast Fourier transform (FFT), but discarded this option because it led to much greater numerical instabilities. It is possible that the running time could instead be improved with the first algorithm in [2], which relies on fast matrix multiplication instead

of FFT. In addition, it should often be possible to exploit independence structure in programs, so the GF can be factored into independent GFs for subsets of the variables. This is not implemented yet because it was not relevant for our benchmarks.

**Performance in practice**   Note that, for many programs, the running time will be better than the worst case. Indeed, substitutions are the most expensive operation, if the substitute expression is of high degree. For some models, like the population model, the substitutions take the form of Theorem C.1, i.e. $G(\boldsymbol{x}[k \mapsto (1-p)x_k])$, which is of degree 1. Such a substitution can be performed in $O(d^n)$ time, because one simply needs to multiply each coefficient by a power of $(1-p)$. This explains why the population model is so fast in the experimental section (Section 5). In other cases as well, we can often make use of the fact that the polynomials have low degree to reduce the space needed to store the coefficients and to speed up the operations.

## C.4   Numerical issues

If our implementation is used in floating-point mode, there is the possibility of rounding errors and numerical instabilities, which could invalidate the results. To enable users to detect this, we implemented an option in our tool to use interval arithmetic in the computation: instead of a single (rounded) number, it keeps track of a lower and upper bound on the correctly rounded result. If this interval is very wide, it indicates numerical instabilities.

Initially, we observed large numerical errors (caused by catastrophic cancellation) for probabilistic programs involving continuous distributions. The reason for this turned out to be the term $\log(x)$ that occurs in the GF of the continuous distributions. The Taylor coefficients of $\log(x)$ at points $0 < z < 1$ are not well-behaved:

$$\log(x) = \log(z) + \sum_{n=1}^{\infty} \frac{(-1)^n z^{-n}}{n}(x-z)^n$$

because the coefficients grow exponentially for $z < 1$ (due to the term $z^{-n}$) and the alternating sign exacerbates this problem by increasing the likelihood of catastrophic cancellations.

To fix this problem, we adopted a slightly modified representation. Instead of computing the Taylor coefficients of the GF $G$ directly, we compute those of the function $H(\boldsymbol{x}) := G(\boldsymbol{x}')$ where $x_i' := x_i$ if $X_i$ is discrete and $x_i' := \exp(x_i)$ if $X_i$ is continuous, thus canceling the logarithm.[6] (The reason we don't use $\boldsymbol{x}' := \exp(\boldsymbol{x})$ for all variables is that for discrete variables, we may need to evaluate at $x_i' = 0$, corresponding to $x_i = -\infty$, which is impossible.) It is straightforward to adapt the GF semantics to this modified representation. Due to the technical nature of the adjustment (case analysis on whether an index corresponds to a discrete variable or not), we do not describe it in detail. With this modified representation, we avoid the catastrophic cancellations and obtain numerically stable results for programs using continuous distributions as well.

# D   Details on the Empirical Evaluation

All experiments were run on a laptop computer with a Intel® Core™ i5-8250U CPU @ 1.60GHz × 8 processor and 16 GiB of RAM, running Ubuntu 22.04.2 and took a few hours to complete overall. The code to run the benchmarks and instructions on how to reproduce our experiments are available in the supplementary material.

## D.1   Comparison with exact inference

We manually patched the tools Dice, Prodigy, and PSI to measure and output the time taken exclusively for inference to ensure a fairer comparison and because the time a tool takes to startup and parsing times are not very interesting from a research perspective. We ran each tool 5 times in a row and took the minimum running time to reduce the effect of noise (garbage collection, background activity on

---

[6]Note that this boils down to using the moment generating function (MGF) $\mathbb{E}[\exp(x_i X_i)]$ for continuous variables $X_i$ and the probability generating function (PGF) $\mathbb{E}[x_i^{X_i}]$ for discrete variables $X_i$. In mathematics, the MGF is more often used for continuous distributions and the PGF more commonly for discrete ones. It is interesting to see that from a numerical perspective, this preference is confirmed.

Table 5: Summary of the benchmarks: $n$ is the number of variables, $o$ is the number of observations, $d$ is the sum of all the observed values, i.e. the total order of derivatives that needs to be computed, $s$ is the number of statements, and $p$ the number of program paths.

| Benchmark | $n$ | $o$ | $d$ | $s$ | $p$ | continuous priors? |
|---|---|---|---|---|---|---|
| population model (Fig. 1a) | 1 | 4 | 254 | 13 | 1 | no |
| population (modified, Fig. 1b) | 1 | 4 | 254 | 21 | 16 | no |
| population (two types, Fig. 1c) | 2 | 8 | 277 | 30 | 1 | no |
| switchpoint (Fig. 2a) | 2 | 109 | 188 | 12433 | 111 | yes |
| mixture model (Figs. 2b and 2c) | 2 | 218 | 188 | 329 | $2^{109} \approx 6 \cdot 10^{32}$ | no |
| HMM (Fig. 2d) | 3 | 60 | 51 | 152 | $2^{30} \approx 10^9$ | no |

the computer). In rational mode, we ran Genfer with the option `--rational` and Dice with the option `-wmc-type 1`. For the "clinicalTrial" benchmark, Genfer required 400-bit floating-point numbers via `--precision 400` because 64-bit numbers did not provide enough accuracy. For better performance, PSI was run with the flag `--dp` on finite discrete benchmarks.

The benchmarks were taken from the PSI paper [9, Table 1], excluding "HIV", "LinearRegression1", "TrueSkill", "AddFun/max", "AddFun/sum", and "LearningGaussian". While Genfer supports continuous priors, the excluded benchmarks use continuous distributions in other ways that are not supported (e.g. in observations). Note that Dice and Prodigy have no support for continuous distributions at all. These benchmarks were translated mostly manually to the tools' respective formats, and we checked that the outputs of the tools are consistent with each other.

## D.2 Comparison with approximate inference

**Experimental setup** As explained in Section 5, we ran several of Anglican's inference algorithms. Each algorithm was run with a sampling budget of 1000 and 10000. The precise settings we used are the following:

- Importance Sampling (IS): has no settings
- Lightweight Metropolis Hastings (LMH): has no settings
- Random-Walk Metropolis Hastings (RMH): has two settings: the probability $\alpha$ of using a local MCMC move and the spread $\sigma$ of the local move. We used the default $\alpha = 0.5, \sigma = 1$ and another setting facilitating local moves with $\alpha = 0.8, \sigma = 2$.
- Sequential Monte Carlo (SMC): number of particles $\in \{1, 1000\}$. The default is 1 and we also picked 1000 to see the effect of a larger number of particles while not increasing the running time too much.
- Particle Gibbs (PGibbs): number of particles $\in \{2, 1000\}$. The default is 2 and we also picked 1000 as the other setting for the same reason as for SMC.
- Interacting Particle MCMC (IPMCMC): it has the settings "number of particles per sweep", "number of nodes running SMC and CSMC", "number of nodes running CSMC", and "whether to return all particles (or just one)". We left the last three settings at their defaults (32 nodes, 16 nodes running CSMC, return all particles). For the number of nodes and the number of nodes we chose the default setting (2) and a higher one (1000) for the same reason as SMC.

A full list of Anglican's inference algorithms and their settings can be found at `https://probprog.github.io/anglican/inference`. Each inference algorithm was run 20 times on each benchmark and the running times were averaged to reduce noise. This took around an hour per benchmark.

## D.3 Benchmarks with infinite support

A summary with important information on each benchmark can be found in Table 5, including some statistics about the probabilistic programs. We describe each benchmark in more detail in the following.

**Population models** The probabilistic program for the original population model from [26] is shown in Fig. 6a. This program can also be written without the extra variable $New$ by writing the two

$$N := \mathsf{Poisson}(\lambda_0);$$

$$N \sim \mathsf{Binomial}(N, \delta);$$
$$New \sim \mathsf{Poisson}(\lambda_1);$$
$$N := New + N;$$
$$\text{observe } y_1 \sim \mathsf{Binomial}(N, \rho);$$

$$N := \mathsf{Poisson}(\lambda_0);$$

$$N \sim \mathsf{Binomial}(N, \delta);$$
$$Disaster \sim \mathsf{Bernoulli}(0.1);$$
```
if Disaster = 1 {
    N +∼ Poisson(λ₁')
} else {
    N +∼ Poisson(λ₁)
}
```
$$\text{observe } y_1 \sim \mathsf{Binomial}(N, \rho);$$

$$N_1 \sim \mathsf{Poisson}(\lambda_0^{(1)});$$
$$N_2 \sim \mathsf{Poisson}(\lambda_0^{(2)});$$

$$N_2 +\sim \mathsf{Binomial}(N_1, \gamma);$$
$$N_1 \sim \mathsf{Binomial}(N_1, \delta_1);$$
$$N_2 \sim \mathsf{Binomial}(N_2, \delta_2);$$
$$N_1 +\sim \mathsf{Poisson}(\lambda_1^1);$$
$$N_2 +\sim \mathsf{Poisson}(\lambda_1^2);$$
$$\text{observe } y_1^{(1)} \sim \mathsf{Binomial}(N_1, \rho);$$
$$\text{observe } y_1^{(2)} \sim \mathsf{Binomial}(N_2, \rho);$$

(a) Original model from [26].   (b) Randomly modified arrival rate.   (c) Two interacting populations.

Figure 6: The program code for the population model variations.

statements involving it as $N +\sim \mathsf{Poisson}(\lambda_1)$. Here we used the same parameter values as [26]: $\delta = 0.2636$, $\lambda = \Lambda \cdot (0.0257, 0.1163, 0.2104, 0.1504, 0.0428)$, and $\Lambda = 2000$ is the population size parameter. Note that the largest population size considered by [26] is $\Lambda = 500$. For the observation rate $\rho$, Winner and Sheldon [26] consider values from 0.05 to 0.95. For simplicity, we set it to $\rho = 0.2$. The $y_k$ are $k = 4$ simulated data points: $(45, 98, 73, 38)$ for $\Lambda = 2000$.

The modified population example is shown in Fig. 6b, where the arrival rate is reduced to $\lambda' := 0.1\lambda$ in the case of a natural disaster, which happens with probability 0.1. The rest of the model is the same.

The model of two interacting populations (multitype branching process) is programmed as shown in Fig. 6c, where $\lambda^{(1)} := 0.9\lambda$, $\lambda^{(2)} := 0.1\lambda$, $\gamma := 0.1$. In this model, some individuals of the first kind (i.e. in $N_1$) can turn into individuals of the second kind and are added to $N_2$. The other parameters of this model are the same as before. The data points are $y^{(1)} = (35, 83, 78, 58)$ and $y^{(2)} = (3, 6, 10, 4)$.

**Bayesian switchpoint analysis**   Bayesian switchpoint analysis is about detecting a change in the frequency of certain events over time. An example is the frequency of coal mining disasters in the United States from 1851 to 1962, as discussed in the PyMC3 tutorial [23]. We use the same model as in [23], and its probabilistic program is shown in Fig. 7a. This situation can be modeled with two Poisson distributions with parameters $\Lambda_1$ (before the change) and $\Lambda_2$ (after the change). Suppose we have observations $y_1, \ldots, y_n$, with $y_t \sim \mathsf{Poisson}(\Lambda_1)$ if $t < T$ and $y_t \sim \mathsf{Poisson}(\Lambda_2)$ if $t \geq T$. The parameters $\Lambda_1, \Lambda_2$ are given exponential priors $\mathsf{Exponential}(1)$. The change point $T$ itself is assumed to be uniformly distributed: $T \sim \mathsf{Uniform}\{1..n\}$. The probabilistic program code is shown in Fig. 7a, where $n = 111$ and $y = (4, 5, 4, 0, 1, 4, 3, 4, 0, 6, 3, 3, 4, 0, 2, 6, 3, 3, 5, 4, 5, 3, 1, 4, 4, 1, 5, 5, 3, 4, 2, 5, 2, 2, 3, 4, 2, 1, 3, \text{n/a}, 2, 1, 1, 1, 1, 3, 0, 0, 1, 0, 1, 1, 0, 0, 3, 1, 0, 3, 2, 2, 0, 1, 1, 1, 0, 1, 0, 1, 0, 0, 0, 2, 1, 0, 0, 0, 1, 1, 0, 2, 3, 3, 1, \text{n/a}, 2, 1, 1, 1, 1, 2, 4, 2, 0, 0, 1, 4, 0, 0, 0, 1, 0, 0, 0, 0, 1, 0, 0, 1, 0, 1)$ where "n/a" indicates missing data (which is omitted in the program).

In fact, this program can be rewritten to the more efficient form shown in Fig. 7b, to minimize the number of variables needed.

**Mixture model**   In the binary mixture model, the frequency events are observed from an equal-weight mixture of two Poisson distributions with different parameters $\Lambda_1, \Lambda_2$, which are in discrete steps of 0.1. This is implemented by imposing a geometric prior and multiplying the rate by 0.1 in the Poisson distribution: $\mathsf{Poisson}(0.1 \cdot \Lambda_1)$. The data is the same as for the switchpoint model. The task is to infer the first rate $\Lambda_1$. The probabilistic program is shown in Fig. 8a.

```
T ∼ Uniform{1..n};
Λ ∼ Exponential(1);
if 1 ∼ Bernoulli(1/n) {
        observe y₁ ∼ Poisson(Λ);
        Λ ∼ Exponential(1);
        observe y₂ ∼ Poisson(Λ);

        ⋮

        observe yₙ ∼ Poisson(Λ);
        T := 1;
} else {if 1 ∼ Bernoulli(1/(n − 1)) {
        observe y₁ ∼ Poisson(Λ);
        observe y₂ ∼ Poisson(Λ);
        Λ ∼ Exponential(1);
        observe y₃ ∼ Poisson(Λ);

        ⋮

        observe yₙ ∼ Poisson(Λ);
        T := 2;
} else {

        ⋱

}}
```

$T \sim \mathsf{Uniform}\{1..n\}$;
$\Lambda \sim \mathsf{Exponential}(1)$;
if $1 \sim \mathsf{Bernoulli}(1/n)$ {
  observe $y_1 \sim \mathsf{Poisson}(\Lambda)$;
  $\Lambda \sim \mathsf{Exponential}(1)$;
  observe $y_2 \sim \mathsf{Poisson}(\Lambda)$;

  $\vdots$

  observe $y_n \sim \mathsf{Poisson}(\Lambda)$;
  $T := 1$;
} else {if $1 \sim \mathsf{Bernoulli}(1/(n-1))$ {
  observe $y_1 \sim \mathsf{Poisson}(\Lambda)$;
  observe $y_2 \sim \mathsf{Poisson}(\Lambda)$;
  $\Lambda \sim \mathsf{Exponential}(1)$;
  observe $y_3 \sim \mathsf{Poisson}(\Lambda)$;

  $\vdots$

  observe $y_n \sim \mathsf{Poisson}(\Lambda)$;
  $T := 2$;
} else {

  $\ddots$

}}

(a) Switchpoint model from [23].

$T \sim \mathsf{Uniform}\{1..n\}$;
$\Lambda_1 \sim \mathsf{Exponential}(1)$;
$\Lambda_2 \sim \mathsf{Exponential}(1)$;
if $1 < T$ {
  observe $y_1 \sim \mathsf{Poisson}(\Lambda_1)$;
} else {
  observe $y_1 \sim \mathsf{Poisson}(\Lambda_2)$;
}

  $\vdots$

if $n < T$ {
  observe $y_n \sim \mathsf{Poisson}(\Lambda_1)$;
} else {
  observe $y_n \sim \mathsf{Poisson}(\Lambda_2)$;
}

(b) More efficient version of the switchpoint model.

Figure 7: The program code for the Bayesian switchpoint analysis.

$\Lambda_1 \sim \mathsf{Geometric}(0.1)$;
$\Lambda_2 \sim \mathsf{Geometric}(0.1)$;
if $1 \sim \mathsf{Bernoulli}(0.5)$ {
  observe $y_1 \sim \mathsf{Poisson}(0.1 \cdot \Lambda_1)$
} else {
  observe $y_1 \sim \mathsf{Poisson}(0.1 \cdot \Lambda_2)$
}

  $\vdots$

if $1 \sim \mathsf{Bernoulli}(0.5)$ {
  observe $y_m \sim \mathsf{Poisson}(0.1 \cdot \Lambda_1)$
} else {
  observe $y_m \sim \mathsf{Poisson}(0.1 \cdot \Lambda_2)$
}

(a) Program code for the mixture model.

$Z := 1$;
$\Lambda_1 \sim \mathsf{Geometric}(0.1)$;
$\Lambda_2 \sim \mathsf{Geometric}(0.1)$;
if $Z = 0$ {
  observe $y_1 \sim \mathsf{Poisson}(0.1 \cdot \Lambda_1)$;
  $Z \sim \mathsf{Bernoulli}(0.2)$;
} else {
  observe $y_1 \sim \mathsf{Poisson}(0.1 \cdot \Lambda_2)$;
  $Z \sim \mathsf{Bernoulli}(0.8)$; }

  $\vdots$

if $Z = 0$ {
  observe $y_m \sim \mathsf{Poisson}(0.1 \cdot \Lambda_1)$;
  $Z \sim \mathsf{Bernoulli}(0.2)$;
} else {
  observe $y_m \sim \mathsf{Poisson}(0.1 \cdot \Lambda_2)$;
  $Z \sim \mathsf{Bernoulli}(0.8)$; }

(b) Program code for the HMM.

Figure 8: The program code for the mixture and HMM model.

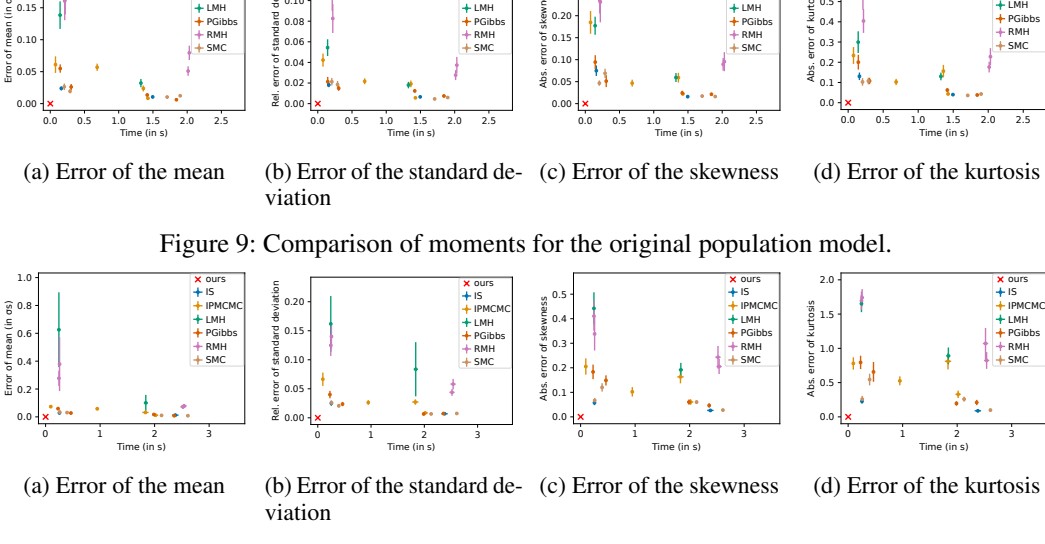

(a) Error of the mean  (b) Error of the standard deviation  (c) Error of the skewness  (d) Error of the kurtosis

Figure 9: Comparison of moments for the original population model.

(a) Error of the mean  (b) Error of the standard deviation  (c) Error of the skewness  (d) Error of the kurtosis

Figure 10: Comparison of moments for the modified population model.

**Hidden Markov model**    The hidden Markov example is based on [22, Section 2.2]. The hidden state $Z$ can be 0 or 1 and transitions to the other state with probability 0.2. The rate of the Poisson-distributed number of observed events depends on the state $Z$ and is $0.1 \cdot \Lambda_1$ or $0.1 \cdot \Lambda_2$, where we impose a geometric prior on $\Lambda_1$ and $\Lambda_2$. As for the mixture model, this discretizes the rates in steps of 0.1. The inference problem is to infer the first rate $\Lambda_1$. The probabilistic program is shown in Fig. 8b, where $y_k$ are $m = 30$ simulated data points from the true rates $\lambda_1 := 0.5$ and $\lambda_2 := 2.5$. Concretely, we have $y = (2, 2, 4, 0, 0, 0, 0, 0, 1, 1, 0, 2, 4, 3, 3, 5, 1, 2, 3, 1, 3, 3, 0, 0, 2, 0, 0, 2, 6, 1)$.

### D.4    Additional results

In the main text, the data we presented was mainly in form of the total variation distance (TVD) between the true posterior (computed by our method) and the approximated posterior (computed by Anglican's inference algorithms). In this section, we present additional evidence in two forms:

- errors of the posterior moments (mean, standard deviation, skewness, and kurtosis) in Figs. 9 to 14, and

- histograms (Fig. 15) of the sampled distribution produced by the best MCMC algorithm, where the sampling budget is picked such that the running time is close to that of our method (e.g. 1000 for the (fast) population model and 10000 for the (slower) mixture model).

The additional plots confirm the conclusions from Section 5.

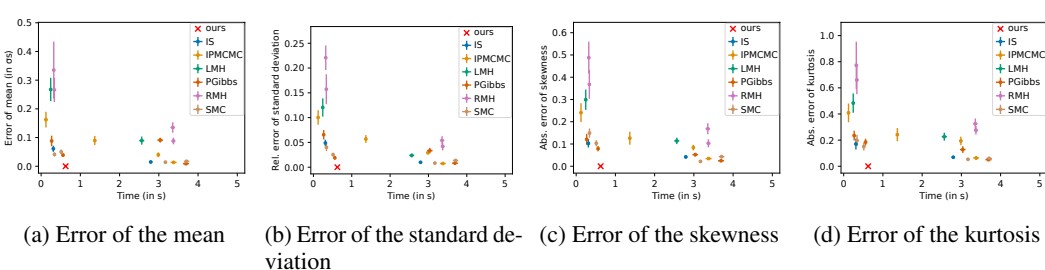

(a) Error of the mean  (b) Error of the standard deviation  (c) Error of the skewness  (d) Error of the kurtosis

Figure 11: Comparison of moments for the two-type population model.

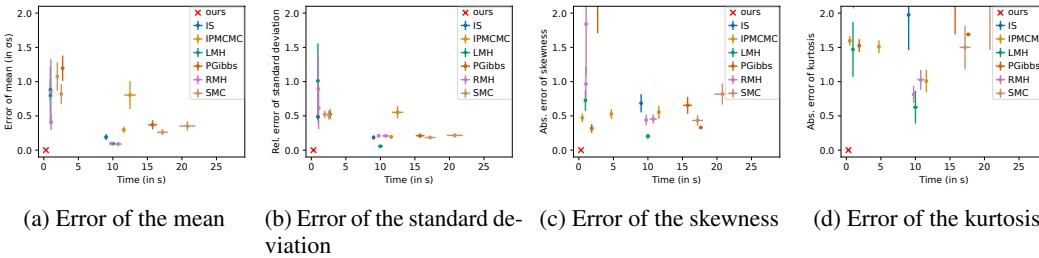

(a) Error of the mean     (b) Error of the standard de-   (c) Error of the skewness    (d) Error of the kurtosis
viation

Figure 12: Comparison of moments for the switchpoint model.

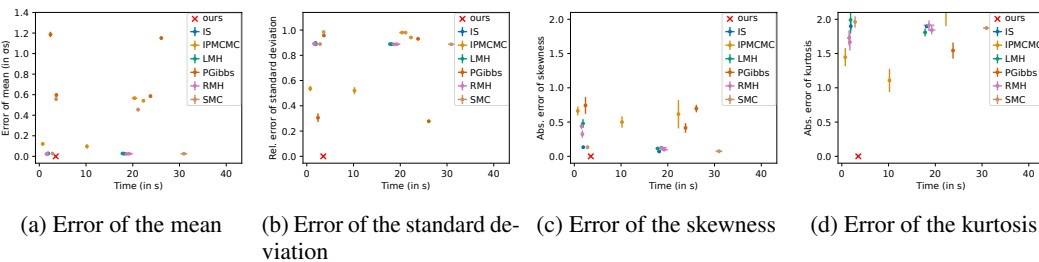

(a) Error of the mean     (b) Error of the standard de-   (c) Error of the skewness    (d) Error of the kurtosis
viation

Figure 13: Comparison of moments for the mixture model.

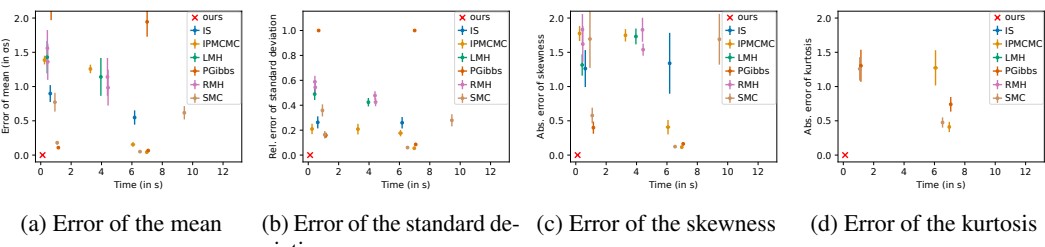

(a) Error of the mean     (b) Error of the standard de-   (c) Error of the skewness    (d) Error of the kurtosis
viation

Figure 14: Comparison of moments for the HMM.

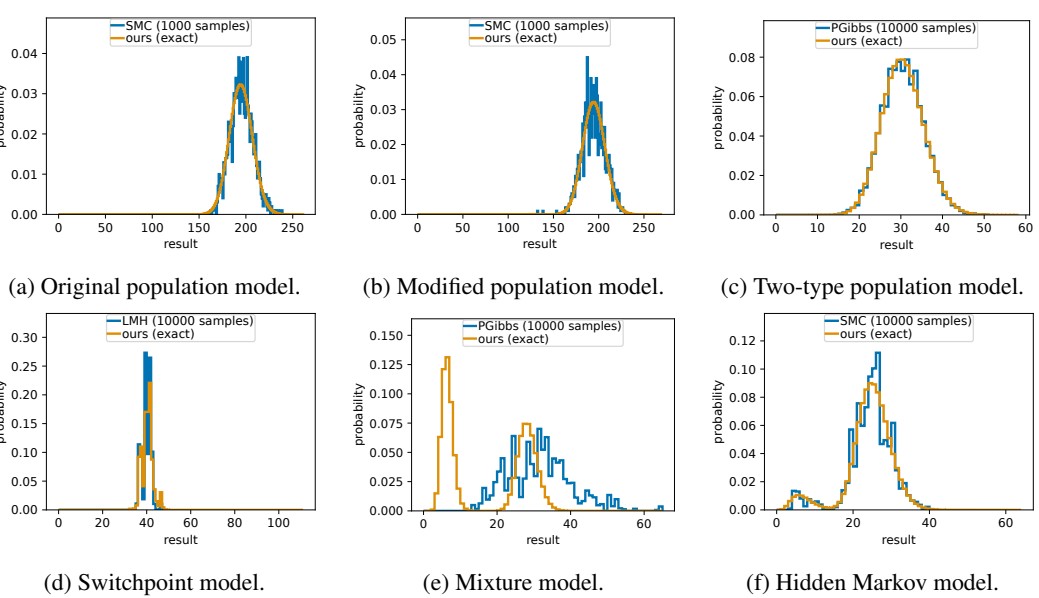

(a) Original population model.    (b) Modified population model.    (c) Two-type population model.

(d) Switchpoint model.      (e) Mixture model.      (f) Hidden Markov model.

Figure 15: Histograms of the exact distribution and the MCMC samples with the lowest TVD and
similar running time to our exact method.

# Contents

