# OpenReview forum: "Exact Bayesian Inference on Discrete Models via Probability Generating Functions: A Probabilistic Programming Approach"
_NeurIPS.cc/2023/Conference — NeurIPS 2023 oral_

### Official Review · Reviewer_JaNH · 2023-07-06

**Soundness:** 4 excellent
**Presentation:** 3 good
**Contribution:** 4 excellent
**Rating:** 8
**Confidence:** 4

**Summary:**

This paper proposes a new approach to exact inference in probabilistic programs. Existing approaches to exact inference attempt to compute marginal and normalized probabilities by efficiently computing the necessary (finite) sums, or using computer algebra to recognize and solve tractable integrals. By contrast, the present work constructs the moment-generating function ($G(x) = \mathbb{E}_{X \sim \mu}[x^X]$) for the distribution represented by the program, and takes its (higher-order) derivatives (using AD) to compute exact posterior marginal probabilities, as well as exact moments. This enables automated exact inference in a broader class of models than handled by existing techniques. The approach is implemented in Rust, and experiments in four example models show that their implementation typically computes exact answers in less time than it takes for Anglican’s generic Monte Carlo algorithms (e.g., sequential Monte Carlo with the prior as the proposal) to converge to accurate estimates.

**Strengths:**

The key strengths I see in this paper are:

- It introduces a significant new approach to PPL inference (automatic differentiation on compositionally constructed moment-generating functions), and I believe it’s important enough that most people working in PPLs will want to understand the new technique. The exposition is clear and should be accessible to a broad probabilistic programming audience.
- The generating function approach resolves a key limitation of other exact inference methods: it can marginalize latent variables with infinite support.
- Limitations of the language, and performance concerns, are clearly outlined (e.g., in the Restrictions paragraph on L147, and the Limitations paragraph on L83).
- The appendix contains rigorous correctness proofs and further exposition of the generating function semantics. There is also a Rust implementation that helps clarify the actual implementation of the technique (e.g., the data representation of Taylor polynomials used by the AD algorithm).

**Weaknesses:**

The key weaknesses I notice are:

- The novel capabilities (and limitations) with respect to the line of work by Klinkenberg et al. could be more clearly described. It would also be good to cite and explain the differences with their recent LAFI abstract, “Exact Probabilistic Inference Using Generating Functions” (https://arxiv.org/pdf/2302.00513.pdf). That workshop abstract is non-archival and much less thorough than the present work, so I don’t mean this as a critique of your work’s novelty. But I think readers will appreciate a clear-eyed description of exactly how their proposed approach differs from yours.
- It is unclear to what extent the limitations of the language are fundamental limitations of the generating function technique, vs. concessions made to get to a simple first publication/implementation of the idea. Could the technique be extended to programs with loops, or does that seem unlikely? What about compound distributions with more than one random parameter (e.g., Binomial where both $n$ and $p$ is also random)? What about ‘score statements’ that multiply a non-negative (affine?) expression into the likelihood? (It seems this should be possible for scores < 1, because we can `observe 1 ~ Bernoulli(X)`, but can it be made to work for more general scores?)
- The experiments are all on programs for which existing exact inference methods fail. But it would be nice to understand how the new approach compares to existing methods on the programs that both approaches can handle. (Even if it is much less efficient, I believe the method still warrants publication, but it would be useful to readers to understand the expressiveness / efficiency trade-offs clearly.)
- L310-312 incorrectly state that WebPPL does not support SMC or particle MCMC. See the SMC and PMCMC implementations here: https://github.com/probmods/webppl/tree/master/src/inference.

**Questions:**

1. Are you 100% sure that systems like Hakaru and $\lambda$PSI cannot handle your examples? I don’t have a great handle on their capabilities, but I expect Maple’s computer algebra simplifications can handle at least some infinite sums and integrals. Have you tried implementing the example programs and seeing what these tools spit out?
2. What are the prospects for a generating-function semantics of a functional (rather than imperative) source language, such as Anglican (but suitably restricted)?

**Limitations:**

The paper is up-front about both the expressiveness limitations of the probabilistic language they support, and the factors in which their algorithm scales exponentially.

---

> ### Author Rebuttal · Authors · 2023-08-09
>
> > The novel capabilities (and limitations) with respect to the line of work by Klinkenberg et al. could be more clearly described. It would also be good to cite and explain the differences with their recent LAFI abstract
>
> The ideas alluded to in their LAFI abstract are now available in a more fleshed-out form as a preprint (https://arxiv.org/abs/2307.07314). (Note that this was posted on arXiv after the NeurIPS submission deadline.) The main difference is that their work only supports discrete distributions and yields a symbolic representation of the GF for the posterior distribution. They also have some support for loops, which requires the user to provide a loop invariant template. On the other hand, our approach allows for some continuous distributions and is (usually) much more performant and scalable, because we only evaluate the derivatives of the generating function instead of representing them fully symbolically (cf. the comparison with PSI and Prodigy in the general rebuttal). We will add a discussion of this to the paper.
>
> > It is unclear to what extent the limitations of the language are fundamental limitations of the generating function technique, vs. concessions made to get to a simple first publication/implementation of the idea.
>
> We give a high-level answer to this in the general rebuttal. Details on your specific questions follow here.
>
> > Could the technique be extended to programs with loops, or does that seem unlikely?
>
> Supporting exact inference for programs with loops seems extremely difficult, unless the inference algorithm is given additional information, e.g. a loop invariant (see the work of Klinkenberg et al.). However, finding such a loop invariant for nontrivial programs seems exceedingly difficult. Furthermore, the variables of looping programs may have infinite expected values, so the generating function may not be definable at the point 1. But in our semantics, evaluating at 1 (for marginalization) is very important. So there are many challenges and opportunities for future research here.
>
> > What about compound distributions with more than one random parameter (e.g., Binomial where both n and p is also random)?
>
> It is hard to say with certainty (somebody might find a closed form that nobody expected), but we tried very hard to find additional compound distributions that admit a closed form and were unsuccessful. The only additional compound distributions we can support are `Binomial(n, X)` and `Gamma(X, a)`, as mentioned in the general rebuttal.
>
> > What about ‘score statements’ that multiply a non-negative (affine?) expression into the likelihood? It seems this should be possible for scores < 1, because we can observe 1 ~ Bernoulli(X), but can it be made to work for more general scores?
>
> As mentioned in the general rebuttal, `score a^X` for nonnegative real `a` < 1 and variable `X` should be supportable. (For `a` > 1, this runs into problems: the GF may not be defined at 1 anymore.) Nonnegative affine scoring (or even nonnegative polynomial scoring) should be possible as well, using a similar technique as for observing from `Bernoulli(X)`. We didn’t include this because we did not find any real-world example to use it.
>
> > But it would be nice to understand how the new approach compares to existing methods on the programs that both approaches can handle.
>
> In the general rebuttal, we attached a comparison with Klinkenberg’s approach to generating functions and the PSI solver.
>
> > L310-312 incorrectly state that WebPPL does not support SMC or particle MCMC
>
> Thank you very much for pointing this out. We will of course correct this statement in the paper. It seems that the documentation of WebPPL’s inference algorithms (http://docs.webppl.org/en/master/inference/methods.html) is incomplete and there are more undocumented algorithms available (such as Particle MCMC).
>
> > Are you 100% sure that systems like Hakaru and λPSI cannot handle your examples?
>
> We tested λPSI and it could not solve any of our examples. We did not test Hakaru because it is quite difficult to get running, requires a Maple license, and benchmarks in the PSI paper indicated that PSI was more performant and able to solve more benchmarks than Hakaru. Even if Hakaru is able to solve small examples, as a fully symbolic tool, it is unlikely to scale to our examples given that even Prodigy (which already exploits the generating function structure) times out.
>
> > What are the prospects for a generating-function semantics of a functional (rather than imperative) source language, such as Anglican (but suitably restricted)?
>
> We believe it should be possible to define a generating-function semantics for a functional language (after all, a functional program can be translated to an imperative one), but it is highly nontrivial: semantics for functional languages usually rely on composition, but the generating functions describe the joint distribution of all variables in the program at any point – we never reason “locally” about only a subset of the variables because this would lose information. So this is an open research question.

---

> > ### Comment · Reviewer_JaNH · 2023-08-21
> >
> > Thank you for this very thorough response. The comparison to PSI and Prodigy makes the value of your novel contributions (in particular the AD-based implementation) even clearer, and accordingly I am raising my score to an 8.
> >
> > I hope that in revision you will find the space to include some of the points from your responses in the paper or appendix. In particular, if you have the space, I'd like to see the ideas from your first two bolded paragraphs in the general response included, as well as the point about why GF semantics for functional languages might be more difficult.

---

### Official Review · Reviewer_SkQ5 · 2023-07-06

**Soundness:** 3 good
**Presentation:** 3 good
**Contribution:** 3 good
**Rating:** 6
**Confidence:** 2

**Summary:**

This paper proposes a new probabilistic programming language called statistical guarded command language (SGCL) and an exact Bayesian inference method for any discrete statistical model that can be expressed in SGCL. Empirical evaluatios on a variety of different discrete probabilistic models demonstrate the effectiveness of the method, especially for challenging distributions with discrete variables that have infinite support.

**Strengths:**

- The paper is well organized and clearly written.
- The proposed method seems novel, and is very useful especially for challenging discrete distributions involving variables with infinite support.


**Weaknesses:**

- The paper seems to miss discussions on probabilistic circuits which is a large body of work concerning probabilistic models with tractable inference.
- Given the limitations of the proposed PPL and its specialized use cases, it would also be helpful to add some discussions on how this can potentially be integrated into/work together with other more general PPLs to expand the applicability of the proposed method.
- The current title and abstract seems slightly misleading, in the sense that the proposed PPL is not generally applicable to any discrete distributions. It would be better to rephrase to make the scope of its applicability clearer, and also emphasize its targeted use case is distributions with variables having infinite support.


**Questions:**

See weaknesses.

**Limitations:**

The authors adequately addressed the limitations.

---

> ### Author Rebuttal · Authors · 2023-08-09
>
> > The paper seems to miss discussions on probabilistic circuits which is a large body of work concerning probabilistic models with tractable inference.
>
> Thank you for pointing this out. Probabilistic circuits (Choi et al., 2020) are indeed a family of probabilistic models that allow for efficient and exact inference, unifying and generalizing many previously introduced tractable probabilistic models (e.g. arithmetic circuits, sum-product networks, and cutset networks). Two exact inference systems that we cite are connected to probabilistic circuits: SPPL (Saad et al., PLDI 2021) extends sum-product networks, and Dice (Holtzen et al., OOPSLA 2020) is based on Weighted Model Counting on binary decision diagrams. After the NeurIPS submission deadline we even learned about work applying generating functions to probabilistic circuits, called “Probabilistic Generating Circuits” (Zhang et al., ICML 2021; Harviainen et al., UAI 2023). It should be noted, however, that the latter circuits only support binary random variables, whereas we support variables with infinite support. We will add a discussion of probabilistic circuits, and probabilistic generating circuits in particular, to the related works section of the paper.
>
> > how this can potentially be integrated into/work together with other more general PPLs
>
> Please refer to the general rebuttal regarding this point.
>
> > The current title and abstract seems slightly misleading, in the sense that the proposed PPL is not generally applicable to any discrete distributions. It would be better to rephrase to make the scope of its applicability clearer, and also emphasize its targeted use case is distributions with variables having infinite support.
>
> It is true that the title is not a complete description of our setting, but we believe the title is quite long already, so we would prefer not to add extra clarifications. We’ve made an effort to be as transparent about the limitations and restrictions about our approach as possible (paragraph “Limitations” in the introduction and “Restrictions” in Section 2). In the abstract, we are explicit about which language constructs are supported, so we did not feel this was misleading. We could add a sentence listing unsupported constructs, but this would just be a negation of the supported constructs (no continuous observations, no non-affine functions, etc.), so it seems redundant. There are subtleties (such as which compound distributions we support), but we believe that this would be too detailed for an abstract and we discuss it in the above paragraphs in the paper. However, we will clarify in the abstract that we only support observing events involving *discrete* variables and that our method is not applicable to all discrete distributions.

---

> > ### Comment · Reviewer_SkQ5 · 2023-08-19
> > **Thanks for the response**
> >
> > I thank the authors for the response which addressed my concerns. Please add the relevant discussions and clarifications as mentioned in the rebuttal.

---

### Official Review · Reviewer_hi49 · 2023-07-07

**Soundness:** 4 excellent
**Presentation:** 3 good
**Contribution:** 4 excellent
**Rating:** 8
**Confidence:** 4

**Summary:**

The paper presents a restricted, first-order probabilistic programming language in which inference for infinite-support latent variables, with continuous or discrete sample spaces, admits a closed form.  For experiments they compare against importance sampling, Lightweight M-H, Random Walk M-H (RMH), sequential Monte Carlo, particle Gibbs sampling, and interacting particle MCMC (IPMCMC) in the Anglican universal PPL.  The authors achieve much lower error than Monte Carlo sampling across experiments, typically with significantly less compute.

**Strengths:**

The paper demonstrates that when latent variables are subject only to affine transformations, a first-order (lacking general recursion) probabilistic programming language can admit tractable exact inference for certain important classes of discrete observations. Even when the resulting exact inference engine takes compute time comparable to the Anglican Monte Carlo inference routines, it tends to run in the middle of the pack while providing much lower approximation error.

**Weaknesses:**

Since the paper focuses on proving and benchmarking something that has an exact answer, it has few weaknesses as such.  It could perhaps have done a better job of making explicit the if-and-only-if conditions that must be met for generating functions to support closed-form inference.

The authors have addressed this concern in the rebuttal.

**Questions:**

How could the exact inference algorithm in this paper be used in a larger probabilistic programming framework, such as when probabilistic circuits admitting exact density calculations are applied as proposals in a broader family of inference algorithms?  When the paper mentions outputting posterior probability masses for certain intervals of integer-valued latent variables, why not also include densities for continuous latent variables?

The authors have addressed these questions in their response.

**Limitations:**

The authors have admitted the limitations of their work, and in exchange for those limitations in expressivity they gain performance in both error and compute time.

---

> ### Author Rebuttal · Authors · 2023-08-09
>
> > making explicit the if-and-only-if conditions that must be met for generating functions to support closed-form inference
>
> Please refer to our general rebuttal for the “only-if” aspect (whether additional constructs could be added to the programming language while still preserving a closed form). For the “if”-direction, the syntax of our language guarantees that the generating function has a closed form.
>
> > How could the exact inference algorithm in this paper be used in a larger probabilistic programming framework?
>
> Please refer to the general rebuttal for this point as well.
>
> > When the paper mentions outputting posterior probability masses for certain intervals of integer-valued latent variables, why not also include densities for continuous latent variables?
>
> While there are mathematical ways of recovering the probability density function from a generating function via an inverse Laplace transform, this cannot be automated in practice because it requires solving integrals, which is intractable in general. So outputting posterior densities is not possible, as far as we know.

---

### Official Review · Reviewer_c91c · 2023-07-07

**Soundness:** 4 excellent
**Presentation:** 4 excellent
**Contribution:** 3 good
**Rating:** 8
**Confidence:** 5

**Summary:**

A probabilistic programming framework for exact inference in discrete probabilistic programs with infinite support is introduced. The framework levegares probability generating functions as a first class attribute of discrete distributions. The probabilistic programming language is formally defined, inference is theoretically analysed for correctness. The performance and accuracy are empirically evaluated and compared with baselines on real-world examples.

**Strengths:**

The paper brings the idea of using GF as a first-class property of a distribution into a practical implementation as a probabilistic programming language. Formal properties of the language and inference are rigorously analysed. The empirical evaluation demonstrates capabilities of the approach, and favorably compares to alternatives. The submission is accompanied by a working and well organized code based.

**Weaknesses:**

The PPL is implemented in Rust, a language which, while it certainly has its merits, less popular for statistical modelling and Bayesian inference, which may make adoption of the approach more difficult. This is a matter of taste though.

**Questions:**

You apparently unroll loops in your benchmarks (I assume because the language does not support loops). What prevents you from implementing restricted loops in the language?

The syntax in the examples is richer than the BNF in the paper. What is '+~' for example? Can you define the full syntax in the paper, at least briefly?

Why rust? Have you considered other options?



**Limitations:**

The paper adequately discusses limitations of the proposed solution.

---

> ### Author Rebuttal · Authors · 2023-08-09
>
> > The PPL is implemented in Rust, a language which, while it certainly has its merits, less popular for statistical modelling and Bayesian inference, which may make adoption of the approach more difficult.
>
> It is correct that Rust is not used in Bayesian inference very much. However, it would not be too difficult to implement, say, a Python API calling Rust under the hood. Many Python libraries work that way (e.g. PyTorch calling C++). Developing the Python interface is mostly an engineering problem, whereas we wanted to focus on the research questions and a proof of concept in our work.
>
> > What prevents you from implementing restricted loops in the language?
>
> There is no particular reason why we couldn’t have implemented bounded loops in our language. But since this does not affect the expressiveness of the language, we preferred the conceptual simplicity of omitting them.
>
> > The syntax in the examples is richer than the BNF in the paper. What is '+~' for example? Can you define the full syntax in the paper, at least briefly?
>
> You are correct that the examples use syntax that is slightly richer than the one provided in the paper. We introduced additional constructs to reduce the number of temporary variables needed; this does not affect the expressivity of the language. There are three additional constructs:
>
> * `X +~ D;` stands for `TMP ~ D; X := X + TMP`
> * `X += a * Y + b;` stands for `TMP := a * Y + b; X := X + TMP;`)
> * `if n ~ D { … } else { … }` stands for `TMP ~ D; if TMP = n { … } else { … }`
>
> We will add an explanation of these constructs to the paper.
>
> > Why Rust? Have you considered other options?
>
> The main reasons were low-level control and performance. The operations on the Taylor polynomials (to evaluate derivatives of the generating function) need to be fast and we have a few optimizations that exploit the structure of GFs arising from probabilistic programs. C or C++ would have satisfied this criterion as well, but Rust’s language features like memory safety, enums (tagged unions), and pattern matching made the implementation a lot more pleasant, robust, and easier to maintain. The first author’s experience with Rust was another contributing factor.

---

> > ### Comment · Reviewer_c91c · 2023-08-12
> >
> > I have read the authors' response and stand by my score. Will be glad to see the paper accepted.

---

### Author Rebuttal · Authors · 2023-08-09

First of all, we would like to thank the reviewers for their thoughtful reviews and interesting questions about our work. We are delighted that the reviewers think that our approach is “significant” (JaNH), “novel and very useful” (SkQ5), that our analysis is “rigorous” (c91c, JaNH), that our implementation compares favorably”in the experimental evaluation (c91c, hi49), and that our presentation is “well-organized and clearly written” (SkQ5). We will try to answer the reviewers’ questions in the following.

In this general rebuttal, we will address points that came up in more than one review or are of interest to more than one reviewer. Specific questions will be discussed in the individual responses.

**Integration with more general probabilistic programming systems (reviewers hi49 & SkQ5)**: Given the restrictions on the programs where our method can be applied, it is a natural question to ask how it could be integrated with a more general probabilistic programming system. We believe this should be possible. For example, in a sampling-based inference algorithm, one could imagine using generating functions to solve subprograms exactly if this is possible (rather than sampling the variables in those subprograms as well). Investigating the details and how well such an approach would work in practice is an interesting research question.

**Conditions for closed-form generating functions (reviewers hi49 & JaNH)**: we were asked for exact conditions under which the generating function of a probabilistic program admits a closed form. The answer is that we tried to make the programming language in the paper (SGCL) as expressive as possible while still maintaining a closed form for the generating function. There are only the following possible additions to the language (that we know of) that preserve closed forms for the GFs but that we didn’t include:

* more distributions with constant parameters: as long as its generating function exists and that function and its derivatives can be evaluated, the distribution can be supported.
* the compound distribution Binomial(n, X), where the success probability is a variable, but this is already expressible as the sum of n independent Bernoulli(X) variables.
* the compound distribution Gamma(X, a), where X is the shape parameter
* the constructs `X mod 2 = 0`, `X := max(X - 1, 0);`, `X := sum_iid(D, Y);` where `D` is a primitive distribution and `X`, `Y` nonnegative integer variables, as mentioned in Klinkenberg et al.’s work
* `score(a^X);` statements for `a` < 1 or `score(q(X));` where `q` is a polynomial with nonnegative coefficients should also be supportable. (A `score` statement multiplies the likelihood by the given expression).

We did not include these constructs because we could not find a real-world example that can use them. It is obviously impossible to say with certainty that no other constructs preserve closed forms (because there are infinitely many possible constructs we might not have thought of). But we tried very hard to express more concepts (e.g. observations from continuous distributions) in terms of generating functions but could not find anything else that works. We will add these remarks to the appendix of our paper.

**Comparison with existing exact tools (reviewer JaNH)**: Reviewer JaNH said it would be interesting to compare our new approach with existing ones on benchmarks that both methods can handle. Accordingly, we decided to compare the PSI tool for exact symbolic inference with our tool on benchmarks that both can handle (i.e. finite discrete distributions). Furthermore, the Prodigy tool by Klinkenberg et al. (which also uses generating functions, but in combination with computer algebra instead of automated differentiation) recently gained the ability to deal with conditioning as well. (A preprint of this work was posted to the arXiv after the NeurIPS submission deadline: https://arxiv.org/abs/2307.07314.) Thus, we decided to compare PSI, Prodigy, and our tool on the subset of Prodigy’s benchmarks that all three tools can handle. The results can be found in the attached PDF. As you can see, our tool is very much competitive and often fastest on these examples. Furthermore, we tried running Prodigy on our benchmarks, but it either does not support them (because of continuous priors), runs out of memory, or times out after 30 min (whereas our tool finishes in seconds). This confirms our claim about the better scalability of our method compared to fully symbolic approaches.

---

> ### Comment · Reviewer_hi49 · 2023-08-19
> **Read the rebuttals**
>
> To the authors,
>
> Thank you for your clear and enlightening rebuttal and for addressing my comments.

---

### Decision · Program_Chairs · 2023-09-21

**Decision:**

Accept (oral)

**Comment:**

This paper introduces probabilistic program inference by means of probability generating functions.
Reviewers agree the analysis is rigorous, and the idea is novel in the probabilistic programming context, even if there are some limitations to the technique. Overall a very solid contribution to the field.